

# Multi-dimensional hydrological-hydraulic model with variational data assimilation for river networks and floodplains

Léo Pujol[1], Pierre-André Garambois[2], and Jérôme Monnier[3,4]

[1]Laboratoire des sciences de l'ingenieur, de l'informatique et de l'imagerie (ICUBE), Fluid Mechanics Team, CNRS, Universite de Strasbourg, France
[2]INRAE (Irstea), Aix Marseille Univ, RECOVER, Aix-en-Provence, France
[3]Institut de Mathematiques de Toulouse (IMT), France
[4]INSA Toulouse, France

**Correspondence:** Léo Pujol (leob.pujol@gmail.com)

**Abstract.** This contribution presents a novel multi-dimensional (multi-D) hydraulic-hydrological numerical model with variational data assimilation capabilities. It allows multi-scale modeling over large domains, combining in situ observations with high-resolution hydro-meteorology and satellite data. The multi-D hydraulic model relies on the 2D shallow water equations solved with a 1D2D adapted single finite volume solver. 1Dlike reaches are built through meshing methods that cause the 2D solver to degenerate into 1D. They are connected to 2D portions that act as local zooms, for modeling complex flow zones such as floodplains and confluences, via 1Dlike-2D interfaces. An existing parsimonious hydrological model, GR4H, is implemented and coupled to the hydraulic model. The forward-inverse multi-D computational model is successfully validated on academic and real cases of increasing complexity, including using the second order scheme version. Assimilating multiple observations of flow signatures leads to accurate inferences of multi-variate and spatially distributed parameters among bathymetry-friction, upstream/lateral hydrographs, and hydrological model parameters. This notably demonstrates the possibility for information feedback towards upstream hydrological catchments, that is backward hydrology. A 1Dlike model of part of the Garonne river is built and accurately reproduces flow lines and propagations of a 2D reference model. A multi-D model of the complex Adour basin network, inflowed by the semi-distributed hydrological model, is built. High resolution flow simulations are obtained on a large domain, including fine zooms on floodplains, with a relatively low computational cost since the network contains mostly 1Dlike reaches. The current work constitutes an upgrade of the Dassflow computational platform. The adjoint of the whole tool chain is obtained by automatic code differentiation.

## 1 Introduction

The accurate estimation of storage and fluxes in surface hydrology is an essential scientific question linked to major socio-economic issues in floods and droughts forecasting, particularly with regards to the ongoing climate change and potential intensification of the water cycle and hydrological hazard (Allen et al. (2019); Iturbide et al. (2020)). In this context, advanced numerical modeling tools are crucially needed to both perform meaningful and detailed representations of basin-scale hydrological processes and provide sensible local forecasts. The quantities of interest range from discharge hydrographs on





upstream ungauged parts of the drainage network to their translation into flow depth, velocities and submersion times on downstream floodplains. This information is difficult to access, especially for floods over large territories. Indeed, given the

complexity of physical processes involved, their limited observability and the resulting hydrological responses, hydrological modeling remains a hard task and internal state-fluxes are generally tinged with uncertainties (Beven (1993); Schuite et al. (2019); Milly (1994)). Moreover, the accuracy of high resolution hydraulic computations may still be affected by complex dynamics with wet-dry fronts, multi-scale and uncertain topography-structures and flow model parameters (e.g. friction), uncertain quantities at open boundaries (upstream inflows but also lateral ones due to sudden local runoff, downstream controls

and backwater effects), internal in/outflows in urban areas, and large computational domains (Monnier et al. (2016)). Thus, integrated hydrological-hydraulic approaches are required (e.g. Nguyen et al. (2016); Hocini et al. (2020)). Such approaches are now enabled by the increasing informative richness of multi-source datasets provided by high resolution hydrometeorology and satellite remote sensing in complement to in situ measurements. Nevertheless, reaching high resolution accuracy and computational efficiency for large scale applications remains a difficult challenge because of multi-scale non-linear hydrodynamic

processes over large computational domains and multiple uncertainty sources.

These uncertainties could be reduced by the optimal combination of models and multi-source datasets, including high resolution maps, spatially sparse in situ flow measurements but also the growing amount of earth observation data provided by new generations of satellites, drones and sensors (e.g. Biancamaria et al. (2016, 2017); Schumann and Domeneghetti (2016) among others). Indeed, remote sensing provides very interesting cartographic observations of the variabilities of worldwide

catchments characteristics (topography, soil occupation, surface moisture, snow cover, ...), as well as an unprecedented and increasing hydraulic visibility over river networks (Garambois et al. (2017); Montazem et al. (2019); Rodríguez et al. (2020)). This growing wealth of multi-sensed information is key to the design and improvement of basin-scale models, as shown for accurate river network 1D hydraulic modeling enabled by recent multi-source altimetric and optical satellite data in Pujol et al. (2020) and in Malou et al. (2021) (see also references therein) or accurate 2D local floodplain models with radar sensed flood-

ing extent (Hostache et al. (2010)). In order to exploit this wealth of hydrological and hydraulic information, the complexity of integrated models and assimilation methods has to be adapted to these data that are both heterogeneous in nature and of varied spatio-temporal resolutions. Moreover, an integrated flood modeling approach should also be computationally efficient in order to be applicable over entire catchments that is large computational domains. This study proposes a new integrated hydrological and multi-dimensional hydraulic modeling approach, based on the accurate and robust 2D hydraulic solver presented in

Monnier et al. (2016). It is capable of multi-variate optimization problems of high dimension using multi-source data.

Cascades of 2D hydrological-hydraulic models have been proposed in recent literature, for inundation mapping at large scales using worldwide DEM (e.g. Grimaldi et al. (2018); Fleischmann et al. (2020); Uhe et al. (2020)) with simplified hydraulic modeling) or at finer scale, e.g. at catchment scale for flash floods in Nguyen et al. (2016); Hocini et al. (2020). In those studies, conceptual hydrological models of upstream-lateral sub-catchments are used to inflow hydraulic models of river net-

work and floodplains in a weak coupling approach, mostly performed via external coupling of numerical models. In Grimaldi et al. (2018); Fleischmann et al. (2020); Uhe et al. (2020), a simple 2D storage cell inundation model obtained from 1D non inertial model (Bates et al. (2010) following Hunter et al. (2008), implemented in LISFLOOD-FP model), enables raster



based inundation modeling over very large domains at relatively low computational cost (see also Fleischmann et al. (2020) for coupling of this non inertial model with the large scale hydrological model MGB Collischonn et al. (2007); Pontes et al.

(2017)). In Hocini et al. (2020), an original 2D hydraulic modeling approach, using "precipiton" for the resolution of the full shallow water model, proposed by Davy et al. (2017), is used to compute steady inundation maps of various return periods at high resolution ($5\,\mathrm{m}$) for river networks and floodplains at catchment scale of several thousands of square kilometers (up to $5050\,\mathrm{km}^2$). In Nguyen et al. (2016), an unsteady full 2D shallow water model (Sanders et al. (2010)) is applied at relatively high resolution ($10$ or $30\,\mathrm{m}$) in the river network and floodplains on a $808\,\mathrm{km}^2$ catchment. Note that sequential data assimilation

methods based on the Kalmann filter have been carried out extensively for mono-variate data assimilation with such models (see e.g. Brêda et al. (2019) with simplified hydraulics in a satellite observability context references therein and Table 1) at varying spatio-temporal resolutions. Current model development strives to propose combinations of high resolution accuracy and fast computation times over large domains and to incorporate multi-source data assimilation methods for large spatially and temporally distributed controls. This paper aims at providing an innovative and effective way to achieve these goals.

In order to combine local accuracy and computational efficiency, the association of full 1D and 2D hydraulic models is an appropriate approach for simulating a basin-scale network in a way that is both practical and adequately accurate. Methods for coupling models of different dimensions have been developed Miglio et al. (2005a, b); Amara et al. (2004), classically using domain decomposition Gervasio et al. (2001), or more recently using local 2D 'zooms' overlapping with the 1D domain, in a variational data assimilation framework Gejadze and Monnier (2007); Marin and Monnier (2009). An iterative coupling

strategy is applied in Barthélémy et al. (2018) between a 1D Mascaret and a 2D Telemac operational model and a sequential data assimilation technique is performed for correcting water levels forecasting. A summary of some established 1D and 2D numerical hydraulic models, external coupling methods and optimization-assimilation methods is presented in Table 1. One can spot DassFlow2D (Monnier et al. (2016)) as the only 2D hydraulic model with a second order solver with accurate wet-dry front treatment, parallel computation and adjoint based variational data assimilation capabilities.

The present study details upgrades to the DassFlow variational data assimilation framework (DassFlow2D-V2, see Monnier et al. (2016)) in the form of a new multi-D hydraulic computational code and an integrated hydrological module, leading to DassFlow2D-V3. The proposed multi-D hydraulic code consists in a single finite volume solver applied to a 2D river network. The network is discretized into "1Dlike" reaches connected to high resolution 2D meshes in a single formulation of the SWE. The resulting product allows building large 1Dlike river networks, connected to fine local zooms. The method can lead to

low computational costs over large networks and local fine scale accuracy at zooms where pertinent. The hydraulic model is coupled with a well-established conceptual hydrological model (GR4H state-space Santos et al. (2018)) in a semi-distributed setup. The variational platform can solve high-dimensional optimization problems with descent algorithms and using gradients computed with the adjoint model obtained via the automatic differentiation tool Tapenade (Hascoet and Pascual (2013)). It enables tackling multi-variate, i.e. with composite control vectors (bathymetry, friction, boundary conditions, hydrological

parameters), data assimilation problems given multi-source dataset, heterogeneous in nature and spatio-temporal resolutions (see e.g. Brisset et al. (2018); Pujol et al. (2020)). This integrated tool chain enables information feedback within the whole



| Platform | Model | Mathematical model | Max order | 1D2D SWE coupling | Parallel computation | DA | Sources available |
|---|---|---|---|---|---|---|---|
| HEC-RAS (15) | 1D-2D | $(A,Q)$ and $(h,u,v)$, both locally non-inertial SWE | 1 | Internal (2 solvers) | No | - | No |
| BreZo (73) | 2D | $(h,u,v)$, porosity | 2 | No | Yes | - | No |
| FullSWOF (25) | 1D and 2D | $(h,u,v)$ for both, full SWE | 1 | No | Yes | - | Yes |
| SW2D-LEMON (77; 40) | 2D | $(h,u,v)$, porosity | 1 | No | No | - | Yes |
| Floodos (24) | 2D | $(h,u,v)$, non-inertial SWE | 1 | No | No | - | Yes |
| b-flood (47) | 2D | $(h,u,v)$, full SWE | 1 | No | Yes | - | Yes |
| Telemac-Mascaret (29; 36) | 1D and 2D | $(A,Q)$ and $(h,u,v)$, full SWE | 1 | External (2 solvers) | Yes | EnKF | Yes |
| LISFLOOD-FP (7) | 1D-2Dlike | $(A,Q)$ non-inertial SWE | 1 | No | Yes | EnKF | Yes |
| DassFlow2D (63) | 2D-1Dlike | $(h,u,v)$, full SWE | 2 | Internal (same solver) | Yes | Var | Yes |
| DassFlow1D (14) | 1D | $(A,Q)$, full SWE | 1 | No | No | Var | Yes |

**Table 1.** Some established freeware hydraulic models. "SWE" stands for Shallow Water Equations. The equations resolved are either formulated in $(A,Q)$ (flow section $\left[\mathrm{m}^2\right]$ and at-a-section discharge $\left[\mathrm{m}^3/s\right]$) or in $(h,u,v)$ (water depth $[\mathrm{m}]$ and 2D depth-integrated flow velocities $[\mathrm{m}/s]$). "Max order" refers to the maximum demonstrated scheme order.

computational domain (basin) and especially from downstream to upstream. The source code and synthetic cases are available upon simple request [1].

The remainder of this article is organized as follows. In Section 2, the modeling hypothesis, the computational resolution and inverse methods are detailed. In Section 3, the multi-D coupling scheme is validated on a series of academic cases and several academic and real-like inference setups are investigated. The study is concluded in Section 4, which also outlines potential applications and improvement perspectives that the proposed method and findings bring.

---

[1] http://www.math.univ-toulouse.fr/DassFlow





## 2 The computational hydrological-hydraulic chain

This section presents the integrated and multi-dimensional hydrological-hydraulic model and the data assimilation approach. The model is designed for simulating spatio-temporal flow variabilities over an entire river network, from upstream hydrological responses to complex flow zones (confluences, multichannels portions, floodplains, ...).

The modeling approach which is detailed below, is based on the following ingredients:

– An integrated multi-D hydraulic model: the 2D shallow water equations (SWE) with finite volume solvers from Monnier
et al. (2016) are applied to "1Dlike"-2D composite meshes of river networks using a numerical flux splitting method and an effective friction power-law depending on flow depth.

– A numerically coupled hydrological model, the widely used GR4 model from Perrin et al. (2003) in its state-space version Santos et al. (2018), for the sake of model differentiability.

– A computational inverse method based on VDA algorithms from Monnier et al. (2016); Brisset et al. (2018); Larnier
et al. (2020) enabling spatially distributed calibration and variational data assimilation with the whole chain.

### 2.1 Multi-D hydraulic-hydrological modeling principle

The flow model consists in a spatially-distributed modeling of hydrological responses coupled to a seamless multi-scale "1Dlike"-2D hydraulic model. The core idea of this work is to apply the 2D SW hydraulic model (Eq. 1) on a multi-D discretization $\mathcal{D}_{hy}$ of the computational domain $\Omega$. The discretization (mesh) $\mathcal{D}_{hy}$ is composed of $N$ mixed unstructured trian-
gular/quadrangular cells with interfaces between 1Dlike and 2D zones (Fig. 3). $\Omega$ is composed of a hydrological domain $\Omega_{rr}$ connected to a hydraulic domain $\Omega_{hy}$ (Fig. 1). $\Gamma_{hy-rr}$ is the border of $\Omega_{hy}$ containing interfaces with $\Omega_{rr}$. The unstructured lattice covering $\Omega$ consists in hydrological units for describing upstream/lateral sub-catchments in $\Omega_{rr}$ and mixed unstructured triangular/quadrangular elements in $\Omega_{hy}$.

### 2.2 Hydraulic module

Numerical hydraulic models describing open channel flows generally rely on the resolution of cross sectionally or depth integrated flow equations, respectively the 1D Saint-Venant or 2D SWE (see e.g. Guinot (2010)). While 1D hydraulic models enable a physically sound representation of river flows variabilities in terms of wetted Section $A$ and discharge $Q$ for instance, 2D hydraulic models in flow depth $h$ and depth integrated velocity $\boldsymbol{u} = (u, v)^T$ enable to tackle more complex flow zones such as confluences/diffluences and floodplain flows. The 2D shallow water model used in the proposed approach is presented here
with the adaptation of the finite volume solver from Monnier et al. (2016) for multi-D modeling. Note that 1D Saint-Venant equations are presented in Appendix B along with their resolution method in DassFlow1D (Brisset et al. (2018); Larnier et al. (2020)) that is used for comparison in this study. Next, this Section presents the hydrological module and the inverse algorithm.





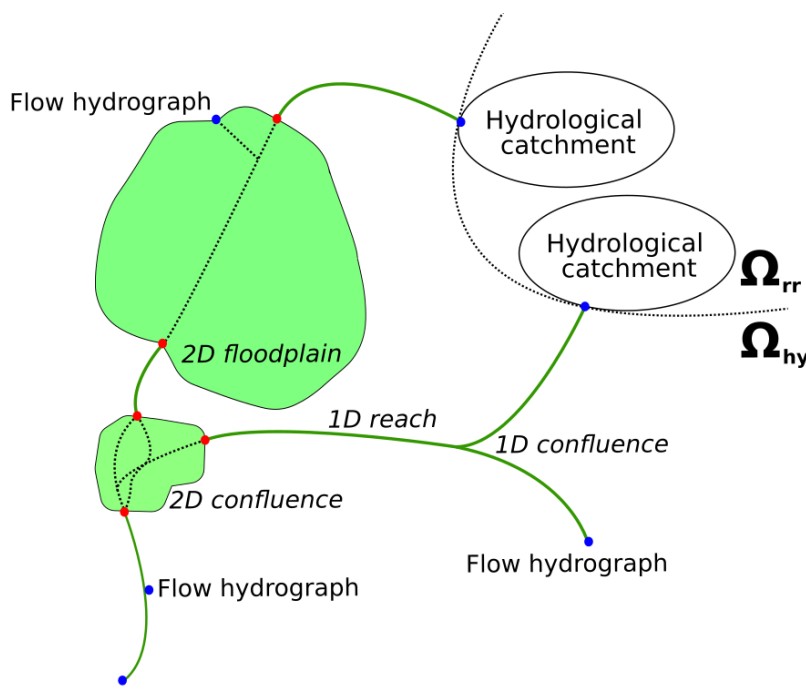

**Figure 1.** Conceptual meshing approach for integrated hydraulic-hydrological and multi-dimensional mod- eling of a river network. The computational domain $\Omega$ is composed of the hydrological domain $\Omega_{rr}$ con- nected to the hydraulic domain $\Omega_{hy}$ . $\Omega_{hy}$ contains 1Dlike meshes and classical 2D meshes, interfaced frontally at the red points. Inflows injected in $\Omega_{hy}$ (blue points) can come from classic inflowing methods or from the coupling to hydrological catchments from $\Omega_{rr}$.

### 2.2.1 Mathematical flow model

On the hydraulic computational domain $\Omega_{hy} \subset \mathbb{R}^2$ and for a time interval $]0, T]$, the 2D SWE with the Manning-Strickler

friction term, in their conservative form, write as follows:

$$\partial_t \mathbf{U} + \partial_x \mathbf{F}(\mathbf{U}) + \partial_y \mathbf{G}(\mathbf{U}) = \mathbf{S}_g(\mathbf{U}) + \mathbf{S}_f(\mathbf{U})$$

$$\mathbf{U} = \begin{bmatrix} h \\ hu \\ hv \end{bmatrix}, \ \mathbf{F}(\mathbf{U}) = \begin{bmatrix} hu \\ hu^2 + \dfrac{gh^2}{2} \\ huv \end{bmatrix}, \ \mathbf{G}(\mathbf{U}) = \begin{bmatrix} hv \\ huv \\ hv^2 + \dfrac{gh^2}{2} \end{bmatrix}, \tag{1}$$

$$\mathbf{S}_g(\mathbf{U}) = \begin{bmatrix} 0 \\ -gh\nabla b \end{bmatrix}, \ \mathbf{S}_f(\mathbf{U}) = \begin{bmatrix} 0 \\ -g\dfrac{n^2 \|\mathbf{u}\|}{h^{1/3}}\mathbf{u} \end{bmatrix}$$





with $h$ the water depth [m] and $\mathbf{u} = (u, v)^T$ the depth-averaged velocity [m/s] being the flow state variables. $g$ is the gravity magnitude $[\mathrm{m/s^2}]$, $b$ the bed elevation [m] and $n$ the Manning-Strickler friction coefficient $[\mathrm{s/m^{1/3}}]$ being the flow model parameters. Classical initial and boundary conditions adapted to real cases are considered (see Monnier et al. (2016, 2019) for

details).

An effective friction law consisting in a simple power-law $n = \alpha h^\beta$ is introduced, as previously done for 1D SWE for effective modeling with simplified multi-channel river geometry in Garambois et al. (2017); Brisset et al. (2018).

### 2.2.2 Building-up equivalencies between 2D and 1D flow states

1Dlike reaches refer to river reaches where the following meshing strategy has been applied: quadrangular cells are built such

that their interfaces are perpendicular to the main flow direction and span the whole river (bankfull) width as a traditional 1D cross-section (XS) would. Examples can be found in Fig. 6 and Fig. 13. This leads to a series of quadrangular cells, each linked to a single upstream and downstream cell. The 1Dlike approach implicitly assumes a rectangular XS shape which potentially impacts the representation of: (i) at a section hydraulic geometry (Leopold and Maddock (1953)), (ii) longitudinal hydraulic controls and flow variabilities.

In view to put the multi-D model in coherence with real flow physics, a continuity condition between 1D and 1Dlike models states and parameters is required. This continuity condition is enforced at-a-section, in a prismatic channel such that the uniform permanent flows, that is equilibrium, are preserved.

Let us consider a reference 1D model in $(A, Q)$ variables with the bankfull width value $W_{1D}$. The friction term reads $S_{f,1D} = \frac{n^2 Q|Q|}{A^2 R_h^{4/3}}$ (Appendix B). In the corresponding 1Dlike model in $(h, u, v)$ variables, see Section 2.2.1, the friction term

reads: $S_{f,1Dlike} = \frac{n^2 ||\boldsymbol{u}||}{h^{1/3}} \boldsymbol{u}$

Considering 1D flow states over an idealized river section (Fig. 2, left) the hypothesis of local flow equilibrium (uniform, steady-state) with identical wetted areas $A$ and WS widths $W$, the continuity condition implies that:

$$n_{1Dlike} = n_{1D} \sqrt{\frac{A}{W} \frac{h_{1Dlike}^{1/3}}{R_{h,1D}^{4/3}}} \qquad (2)$$

where $n_{1Dlike}$ (resp. $h_{1Dlike}$) is the Manning-Strickler friction coefficient (resp. flow depth) in the 1Dlike model i.e. the

coefficient in the 2D SWE (Eq. 1).

With the additional assumption of a rectangular XS (as it will be assumed in some test cases), we have $h_{1Dlike} = A/W_{1D}$, which leads to $n_{1Dlike} = n_{1D} \left( \frac{h_{1D}}{R_{h,1D}} \right)^{2/3}$.

This "1Dlike equivalent friction" leads to a perfect fit in WS elevation of 1Dlike model and 1D model in a straight prismatic channel at the given uniform regime (results not shown here for brevity). Fig. 2, right, shows the evolution of the ratio $h/R_h$

vs $h$. For rectangular and parabolic XS, the ratio $n_{1Dlike}/n_{1D}$ is expected to increase with $h$. Thus, one can naturally expect an overestimation (resp. underestimation) of the actual friction coefficient by $n_{1Dlike}$ at lower flows (resp. greater flows). This would lead to an overestimation (resp. underestimation) of the 1D WS elevation by the "equivalent" 1Dlike model. However,





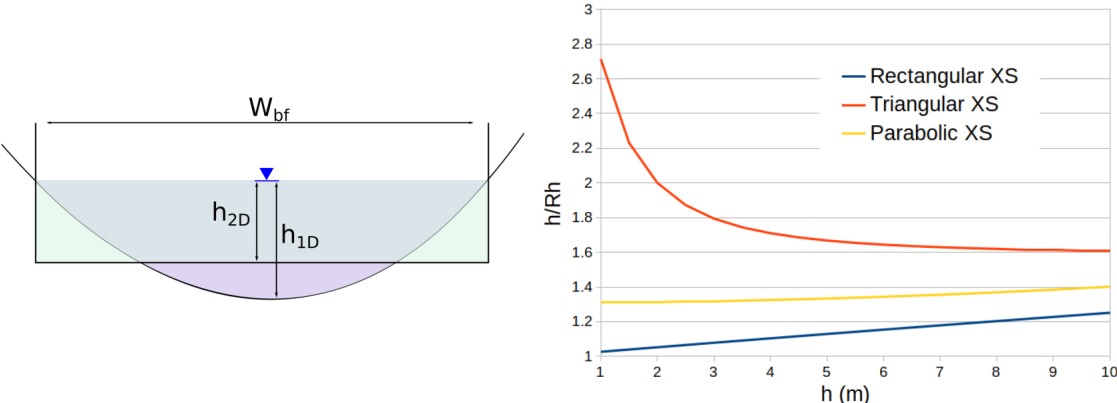

**Figure 2.** Equivalency of 1D and 2D flow states at equilibrium (permanent uniform flows): effective friction and bathymetry. Left: Equivalency between a 1D idealized XS (purple) and a 2D single-cell rectangular XS (green), with the same flow section $A$ and WS elevation. Right: Variation of the hydraulic radius $R_h(h)$ for 3 XS shapes (of similar dimensions). This showcases the potential over- and under-estimation of state variables using "1Dlike equivalent friction" from Eq. (2).

later it will be considered a power-law in $h$ to model the friction coefficient (Subsection 2.2.1), which provides a better fit to the 1D WS elevations outside of the considered permanent flow.

Note that longitudinal controls and flow variabilities in 1Dlike models are assessed using synthetic cases in Subsection 3.2.2.

### 2.2.3    Multi-dimensional hydraulic model

Over a given cell $K \in \Omega_{hy}$ of area $m_K$, the piecewise constant values $\mathbf{U}_K = \frac{1}{m_K}\int_K \mathbf{U}dK$ are approximated. Recall that the finite volume approach applied to the homogeneous part of the hyperbolic system of Eq. (1) (that is without the friction source term $\mathbf{S}_f$ but including a consistent discretization of the gravitational source term $\mathbf{S}_g$) writes as follows:

$$\bar{\mathbf{U}}_K^{n+1} = \mathbf{U}_K^n - \frac{\Delta t^n}{m_K}\sum_{e \in \partial K} m_e \mathbf{F}_e\left(\mathbf{U}_{K,i}^n, \mathbf{U}_{K_e}^n, \mathbf{n}_{e_i,K}\right) \tag{3}$$

where $\mathbf{U}_K^n$ and $\mathbf{U}_K^{n+1}$ are the piecewise constant approximations of $\mathbf{U} = (h, hu, hv)^T$ at time $t^n$ and $t^{n+1}$ (with $t^{n+1} = t^n + \Delta t^n$), $\mathbf{F}_e$ stands for Riemann fluxes through each edge $e$ of the border $\partial K$ of the cell $K$, with each adjacent cell $K_e$. The length of edge $e$ is $m_e$ and $\mathbf{n}_{e,K}$ is the unit normal to $e$ oriented from $K$ to $K_e$.

The finite volume schemes are those developed in Couderc et al. (2013); Monnier et al. (2016).The discretization of the
friction source term is described in Appendix A.



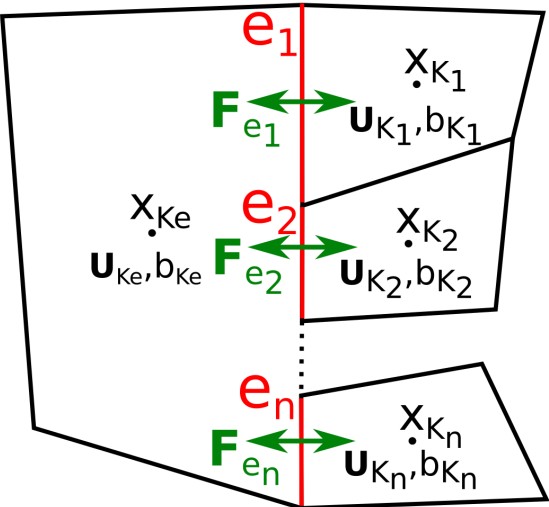

**Figure 3.** Internal multi-D domains interface, general case. At each cell center $\boldsymbol{x}$, the state variables $\mathbf{U} = (h, hu, hv)^T$ and the bathymetry $b$ are defined. The total numerical flux is conserved: $\mathbf{F}_e = \sum\limits_{i=1..n} m_{e_i} \mathbf{F}_{e_i} \left( \mathbf{U}_{K,i}^n, \mathbf{U}_{Ke}^n, \mathbf{n}_{e_i,K} \right)$.

### 2.2.4

Based on the first and second order finite volume solvers of Couderc et al. (2013); Monnier et al. (2016), a 1D2D coupling technique is introduced following a similar concept to Finaud-Guyot et al. (2018) (urban geometries and porosity context) to compute numerical fluxes on each interface between a 1Dlike quadrangular mesh cell connected to several 2D cells as 180 schematized in Fig. 3.

At the multi-D interfaces, that is in the case of $n > 1$ cells $x_{K,i}$ $i \in [1..n]$ adjacent to the same interface of another cell $Ke$ (see Fig. 3 for notations and e.g. Fig. 16 for a real-like example), a special treatment is applied. It consists in the Riemann fluxes being calculated for each cell $K_i$ using the state from the same corresponding $Ke$ cell over an interface of length $m_{e_i}$. In the end, the flux crossing the interface $e = \cup e_i$ is equal to the sum of the fluxes crossing the $e_i$ interfaces: $\mathbf{F}_e = $ 185 $\sum\limits_{i=1..n} m_{e_i} \mathbf{F}_{e_i} \left( \mathbf{U}_{K_i}^n, \mathbf{U}_{Ke}^n, \mathbf{n}_{e_i,K} \right)$.

This type of internal interface has been implemented in the numerical solvers from Monnier et al. (2016) in the DassFlow platform which includes a solver with second order accuracy in space. This solver, developed in Couderc et al. (2013); Monnier et al. (2016), is accurate and robust for wet-dry front propagations and fully applies in the present context. Note that the lateral distribution of variables accross the 1D2D interface is not constrained. The source code and synthetic cases are available upon 190 simple request.



## 2.3 Hydrological module

In order to simulate the hydrological response of sub-catchments within a river basin, a hydrological module is coupled to the 2D SW flow model. The widely used, parsimonious and robust conceptual hydrological model GR4 (Perrin et al. (2003)), in its "state-space" version from Santos et al. (2018), was chosen. The original lumped hydrological model has been deployed in a semi-distributed manner in the DassFlow framework.

The model is composed of two non-linear stores for production (soil moisture accounting) and routing, and a Nash cascade composed of a series of linear stores replacing the unit hydrograph from Perrin et al. (2003). Being a set of ODE with explicit dependency to parameters, this hydrological model is differentiable. Moreover, the Fortran code is differentiable with the automatic differentiation tool Tapenade Hascoet and Pascual (2013).

Let us consider a sub-catchment $bv_i$ among sub-catchments $\{bv_1, ..., bv_C\}$ in the discretized river basin hydrological domain $\Omega_{rr}$. The hydrological model can be seen as a dynamic operator $\mathcal{M}$ relating sub-catchment state variables vector $h(x_i, t)$ with observable input of spatially averaged (on sub-catchment $x_i$) rainfall $P(x_i, t)$ and potential evapotranspiration $E(x_i, t)$, observable outputs $Y(x, t)$ and "unobservable" parameters $\theta(x)$.

Omitting the sub-catchment index $i$ for readability, the hydrological model consists in the following set of ordinary differential equations:

$$\frac{\mathrm{d}\boldsymbol{h}}{\mathrm{d}t} = \begin{cases} \dot{h_p} & = P_s - E_s - P_{erc} \\ \dot{h_1} & = P_r - Q_{Sh,1} \\ \dot{h_2} & = Q_{Sh,1} - Q_{Sh,2} \\ ... & ... \\ \dot{h_{nres}} & = Q_{Sh,nres-1} - Q_{Sh,nres} \\ \dot{h_r} & = Q_9 + F - Q_r \end{cases} \tag{4}$$

where $P_s$, $E_s$, $P_r$, $Q_{Sh,i}, i \in [1..nres]$, $Q_9$, $F$, $Q_r$ are model internal fluxes given in Appendix C along with their internal parameters. The evolution of reservoir states and model outputs and inputs is presented for a sample rain event in Fig. 4.

The input of the hydrological model are the evapotranspiration and precipitation $E_n$ and $P_n$, the output is discharge $q(t) = Q_r + Q_d$ [mm/h]. $E_n$ and $P_n$ are classically imposed from data time series as piecewise constant on fixed temporal resolution (e.g. hourly). The numerical resolution is achieved with an implicit Euler algorithm with an adaptative sub-step algorithm enabling to reduce numerical errors especially for high flows (Santos et al. (2018)). The initial states of the stores is given by a 1 year warm-up run. The discharge $q$ is injected into $\Omega_{hy}$ at a sub-interface of $\Gamma_{hy-rr}$, either as an upstream or lateral flow.

The calibrated parameters will be the classical 4 parameters $(c_i)_{i \in 1..4}$ of GR4 (they will constitute the control vector considered in the forthcoming VDA experiments). Other parameters, such as several drainage law exponents or the number of Nash cascade stores are not optimized in this study. They are set at values from Santos et al. (2018).



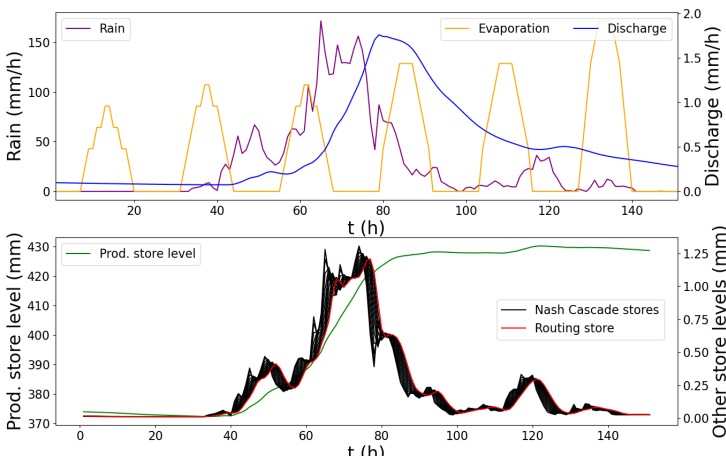

**Figure 4.** Evolution of GR4 inputs, output and reservoirs states during a sample rain event. Top: temporal forcings (rain and evaporation) and modeled output (discharge). Bottom: hydrological model states (reservoir levels).

## 2.4 Inverse algorithm: Variational Data Assimilation

Given spatio-temporal flow observables, provided by in situ and airborne sensors for instance, the inverse algorithm consisting
in Variational Data Assimilation (VDA) aims at estimating the unknown or uncertain "input parameters" of the hydrological-hydraulic chain composed of a hydraulic model, presented in Section 2.2, and a hydrological model, presented in Section 2.3.

We consider the following set of spatio-temporal observations of water surface and discharge over the river domain $\Omega \subset \mathbb{R}^2$:

$$Z_{o,k}(t), \forall k \in [1..N_{o,Z}], Q_{o,k}(t), \forall k \in [1..N_{o,Q}] \tag{5}$$

with $Z_o$ the observed WS elevation [m] above reference elevation, $Q_o$ the observed discharge $[\mathrm{m}^3/\mathrm{s}]$, $N_{o,Z}$ the number of altimetric observations points and $N_{o,Q}$ the number of observed discharges over $\Omega$.

Note that observed discharge $Q_o$ may be a value of a hydraulic discharge at a flow XS in $\Omega_{hy}$, or within the hydrological domain $\Omega_{rr}$ and especially at the outlet of a sub-catchment here. Also note that other water surface observables could be considered, such as water surface velocity observations or dynamic water masks - not the scope of this study.

Given river stage and/or discharge observations, the aim is here to estimate unknown or uncertain quantities of the hydrological-hydraulic model among: discharge hydrographs $Q_i(t), i \in [1, N]$ on the border of the hydraulic domain, spatially distributed hydraulic parameters (bathymetry elevation $b$ or friction $n$) and hydrological models parameter sets $(c_i)_{i \in 1..4}$.

The control vector containing the sought quantities is denoted $\theta$ in what follows:



$$\theta = (\theta_{hy}, \theta_{rr}) = ((Q_1^0, ..., Q_1^T, ..., Q_N^1, ..., Q_N^T, n_1, ..., n_M, b_1, ..., b_M), (c_1, ..., c_P)) \tag{6}$$

with $\theta_{hy}$ and $\theta_{rr}$ the control vectors of respectively the hydraulic and the hydrological modules, $N$ the number of inflows points and $T$ the number of inflow values in time, $M$ the number of modeled cells and $P$ the number of hydrological units.

We consider a cost function $j_{obs}$ aiming at measuring the discrepancy between simulated and observed flow quantities on the computational domain $\Omega$. This cost function is defined as:

$$j_{obs}(\theta) = j_Q(\theta) \ or \ j_{obs}(\theta) = j_Z(\theta) \tag{7}$$

This cost function contains either misfit to WS elevation, $j_Z(\theta) = \frac{1}{2}\|Z_o(t) - Z(\theta,t)\|^2_{\mathcal{O}_{\mathcal{Z}}}$, or misfit to discharge, $j_Q(\theta) = \frac{1}{2}\|Q_o(t) - Q(\theta,t)\|^2_{\mathcal{O}_{\mathcal{Q}}}$.

The metrics (symmetric positive definite matrices) $\mathcal{O}_{\mathcal{Z}}$ and $\mathcal{O}_{\mathcal{Q}}$ are based on the inverse of the observation error covariances. This enables to regularize the inverse problem, see e.g. Bouttier and Courtier (2002); Monnier (2021); Asch et al. (2016) and references therein for related discussions.

Moreover, we classically enrich the cost function with a regularization term: $j(\theta) = j_{obs}(\theta) + \gamma j_{reg}(\theta)$ with $j_{reg}$ a Tikhonov type regularization term. Here we consider a regularization on the bathymetry only: $j_{reg}(\theta) = \frac{1}{2}\sum_{i=1}^{M}\left((\partial_x b_i)^2 + (\partial_y b_i)^2\right)$ with $\theta$ defined by Eq. (6).

The regularization term adds convexity to the cost function. Moreover, it here dampens the bathymetry highest frequencies. Recall that low Froude flows i.e. subcritical flows naturally act as a low pass filtering of the bathymetry shape, see Martin and Monnier (2015); Gudmundsson (2003).

The total cost function $j$ is minimized starting from a background value $\theta^{(0)}$. Following Lorenc et al. (2000), see also Larnier et al. (2020), the following change of variables is applied:

$$k = B^{-1/2}\left(\theta - \theta^{(0)}\right) \tag{8}$$

with $B$ the covariance matrix of the background error.

Then by setting $J(k) = j(\theta)$, the optimization problem which is solved is actually the following:

$$\min_k J(k) \tag{9}$$

The first order optimality condition of this optimization problem (Eq. 9) reads $B^{1/2}\nabla j(\theta) = 0$. The change of variables based on the covariance matrix $B$ acts as a preconditioning of the optimization problem. This optimization problem is solved using a first order gradient-based algorithm, more precisely the classical L-BFGS quasi-Newton algorithm (limited-memory Broyden-Fletcher-Goldfarb-Shanno bound-constrained, Zhu et al. (1997)) or, in some cases in this study, its bounded version L-BFGS-B (Zhu et al. (1997)) but without variable change (Eq. 8). Detailed know-hows on VDA may be found e.g. in online courses (see e.g. Bouttier and Courtier (2002); Monnier (2014)). The gradient is computed with the help of the adjoint model. The latter is obtained by automatic differentiation, using Tapenade Hascoet and Pascual (2013).



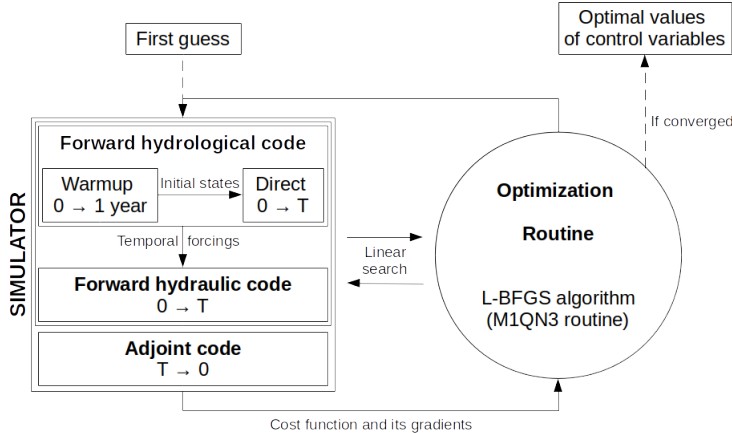

**Figure 5.** Complete VDA hydraulic-hydrological tool chain

The choice of the covariance matrix $B$, represents an important a priori and greatly influences the computed solution of the inverse problem. Assuming the unknown parameters are independent variables, the matrix $B$ is defined as a block diagonal matrix:

$$B = blockdiag\left(B_{\Omega_{hy}}, B_{\Omega_{rr}}\right), with\, B_{\Omega_{hy}} = blockdiag\left(B_Q, B_n, B_{z_b}\right)$$

Each block matrix of $B_{\Omega_{hy}}$ is defined as a covariance matrix (positive definite matrix) using 2D kernels for spatial controls. Here following Larnier et al. (2020), we consider:

$$(B_Q)_{i,j} = (\sigma_Q)^2\, exp\left(-\frac{|t_j - t_i|}{L_Q}\right) \tag{10}$$

$$(B_b)_{i,j} = (\sigma_b)^2\, exp\left(-\frac{|x_j - x_i| + |y_j - y_i|}{L_b}\right), (B_n)_{i,j} = (\sigma_n)^2\, exp\left(-\frac{|x_j - x_i| + |y_j - y_i|}{L_n}\right) \tag{11}$$

The parameters $L_Q$ and $(L_b, L_n)$ act as correlation lengths. These parameters are usually empirically defined. However the expression of $B_b$ and the correlation lengths can be derived from physically-based estimations following Malou and Monnier (2021).

The following stopping criterion are used to stop the iterative optimization process if: i) the cost function does not decrease over a set number of iterations or ii) the current cost gradient, normalized by the initial cost gradient, goes under a set objective value. The multi-D hydrological-hydraulic tool chain presented above has been implemented in the latest version of Monnier et al. (2019).



## 3  Results and discussion

### 3.1  Numerical experiments design

Both synthetic and real cases are considered to test the forward and inverse modeling capabilities of the proposed computational chain for river networks and floodplains simulation. First, the multi-D hydraulic model (Section 2.2.3) is validated against reference hydraulic models on synthetic cases corresponding to typical hydraulic complexities: (i) simple straight channel, (ii) confluence and (iii) straight, rectangular and parabolic, channels with effective parameterization of friction and bathymetry. For fluvial regimes in the context of altimetry, these hydraulic complexities generate hydraulic controls. Following Montazem et al. (2019), we define hydraulic controls as characterized by a maximal deviation of the water depth from the normal depth (see e.g. Chow (1959); Dingman (2009) for definition) at the reach scale. Next, a series of inference cases are considered in twin experiment setups. In a twin experiment, a reference model acts as a synthetic truth and is used to generate observations of model variables (e.g. $b + h$ in $\Omega_{hy}$ or $Q$ in $\Omega_{rr}$). No observation noise is considered in this study. The VDA method is then applied with these observations on an altered version of the reference model. Inferences of temporal forcings (inflow hydrographs $Q_i(t), i \in [1..N]$) are presented on a 1D2D confluence case. Inferences of channels parameters are presented on a straight 1Dlike case. Next, inferences of hydrological parameters $(c_i)_{i \in 1..4}$ are presented using hydraulic observables. Finally, the model is tested on two real cases: (i) the capability of the 1Dlike model to reproduce real flow lines and propagations, through effective bathymetry-friction $(b(x), n(x))$ inference, is assessed against a reference 2D model built on fine bathymetry of $75\,\mathrm{km}$ of the Garonne river and (ii) the whole multi-D hydraulic hydrological tool chain is tested at basin scale on the real complex case of the Adour River network.

### 3.2  Synthetic cases

First, the proposed multi-D hydraulic solver (Subsection 2.2.3) is evaluated and compared to a fine 2D reference model on two simple configurations, a straight channel and a confluence, that feature contain frontal interfaces between 1Dlike and 2D meshes. Next, to investigate the reproductibility of hydraulic controls using a 1Dlike meshing approach and effective modeling, this approach is compared to a 1D reference model in three typical channel configurations: (i) a rectangular prismatic channel, (ii) a rectangular channel with a slope break and (iii) a parabolic prismatic channel. Finally, inferences of inflow hydrographs $Q_i(t), i \in [1..N]$ and of a friction power-law $n = \alpha h^\beta$ are carried out using WS observables.

#### 3.2.1  Forward multi-D hydraulic cases

**Straight channel case**

A prismatic rectangular channel and a multi-D mesh are considered (1Dlike to 2D to 1Dlike, see Fig. 6). The channel width is $300\,\mathrm{m}$ and its length is $2300\,\mathrm{m}$. A rating curve is imposed downstream. This multi-D model is compared to a reference 2D model with refined mesh (mesh not shown, 2400 cells, average edge length $\smile 25\,\mathrm{m}$). The multi-D waterline is validated against the 2D model at permanent flow $\left(Q = 100\,\mathrm{m^3/s}\right)$ and the modeled downstream discharges are compared for a flood





hydrograph (Fig. 6(b)). Both first (not shown) and second order (Fig. 6) numerical solver allow a close fit to the target water line (Fig. 6, bottom).

At permanent flow, a slight misfit is observed between the 2D and multi-D WS elevations with the second order scheme (Fig. 6, top, relative misfit $< 0.15\%$ at $1000\,\mathrm{m}$). This is due to approximation error in the multi-D model caused by large spatial steps in 1Dlike reaches ($dx = 200\,\mathrm{m}$). Indeed, this misfit is reduced by reducing the 1Dlike cells length (not shown). This is

confirmed and showcased in the next subsection.

At the interface between 1Dlike and 2D meshes, a slight jump in WS elevations can be observed at all 2D cells (Fig. 6, top). This is due to the second order scheme, which is currently not designed for multi-D interfaces. Recall that no constraint is imposed on the lateral distribution of computed variables. The technical implementation of this reconstruction for 1D2D interfaces will be done in next version of DassFlow.

During a varied flow event, the outflow of both models is close to identical (Fig. 6, middle). The flow is correctly transmitted at multi-D interfaces and, at this scale, the 1Dlike meshes are adequate to model a flood wave propagation.

**Simple confluence case**

A simple symmetrical confluence is modeled using a multi-D mesh. The channel width is $300\,\mathrm{m}$ in the downstream reach and

$150\,\mathrm{m}$ in the upstream reaches. Two inflow hydrographs are imposed at the two upstream interfaces. The maximum abscissa of the mesh points are 0 and $2075\,\mathrm{m}$. The 2D part (average edge length $\smile 5\,\mathrm{m}$) contains a confluence flow zone (Fig. 6) while the 1Dlike mesh ($dx = 100\,\mathrm{m}$) covers the upstream and downstream reaches. At permanent flow, the multi-D model compares well with the reference 2D fine model (mesh not shown, $15\,000$ cells, average edge length $\smile 5\,\mathrm{m}$) and leads to similar conclusions to that of the above paragraphs. Using a shorter spatial step in the 1Dlike reaches ($10\,\mathrm{m}$) reduces the difference between reference

and multi-D model and allows for a nearly perfect fit to the reference WS elevation at permanent flow (Fig. 6, in green). During a varied flow event, the outflows modeled with the multi-D model are very close to reference ones - slight differences during rising and falling limbs, same peak time, $NSE = 0.996$.

### 3.2.2 Hydraulic controls and effective friction

To investigate the reproductibility of hydraulic controls for fluvial flows, which are characterized by a maximal deviation of

the water depth from the normal depth at the reach scale (see definition in Montazem et al. (2019)), using a 1Dlike approach, a set of typical channel variabilities and hydraulic controls are considered. Let us compare 1D and 1Dlike waterlines in a series of synthetic cases: (i) a straight rectangular channel, (ii) a straight rectangular channel with a mid-channel slope break and (iii) a straight parabolic channel.

**Direct calibration**

Recall the equivalent friction formulation (Eq. 2) designed to make match the water line at-a-section and at equilibrium. For each 1Dlike case, waterlines with 1D friction ($n = 0.05\,\mathrm{s/m^{1/3}}$) and effective equivalent friction are generated. Effective friction values are calculated using (Eq. 2) and results from the corresponding 1D case. Reference 1D flow lines are computed





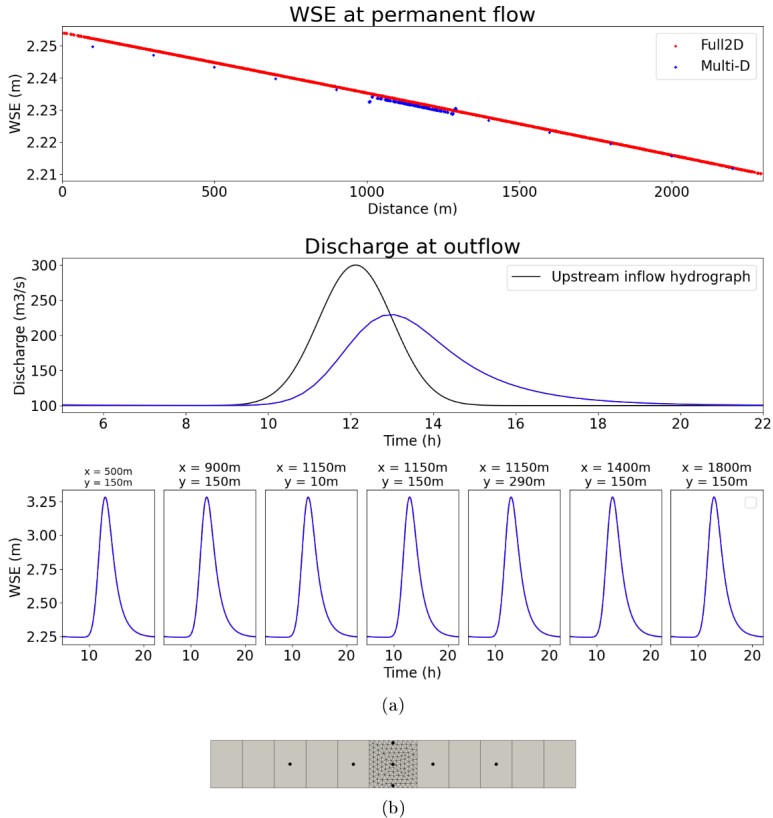

(a)

(b)

**Figure 6.** Multi-D straight channel case results with second order scheme. The flow is subcritical with a maximum Froude value of $0.06$. (a) Top: Waterlines for a permanent flow of $100\,\mathrm{m}^3/\mathrm{s}$. In red: reference 2D waterline. In blue: waterline for the mesh (b). The total misfit at the 1D2D upstream interface (at $1000\,\mathrm{m}$) is around $10^{-3}\,\mathrm{m}$, for a relative misfit $< 0.15\%$ of the local depth. Middle: In black: upstream discharge during sample varied flow event. In red/blue (total overlap): downstream simulated flow for the considered meshes. Bottom: WS elevation observations for the varied flow event (total overlap). (b) Multi-D mesh. Station locations are noted as black dots.

with DassFlow solving 1D Saint-Venant equations in $(A, Q)$ state variables with a Preissmann scheme (see Appendix B and also Larnier et al. (2020)).

In a straight rectangular prismatic channel, with the 1D normal water depth imposed downstream (Fig. 8(a)), a backwater curve is computed by the DassFlow1D model (in red). With a 1D homogeneous friction ($n = 0.05\,\mathrm{s/m}^{1/3}$), the 1Dlike approach yields an underestimated waterline (in blue). An homogeneous effective friction allows to correct the underestimation and to better match the 1D normal water depth over the whole domain (in green). The remaining misfit can be attributed to numerical errors, especially numerical diffusion from the Preissmann scheme (1D model spatial step is $dx_{1D} = 100\,\mathrm{m}$).

A first complexification of this case consists in the introduction of a local hydraulic control point in the form of a slope break at $x = 10\,\mathrm{km}$ (Fig. 8(b)). In this setup, both 1D and 1Dlike models generate M2 backwater curves, see e.g. Dingman

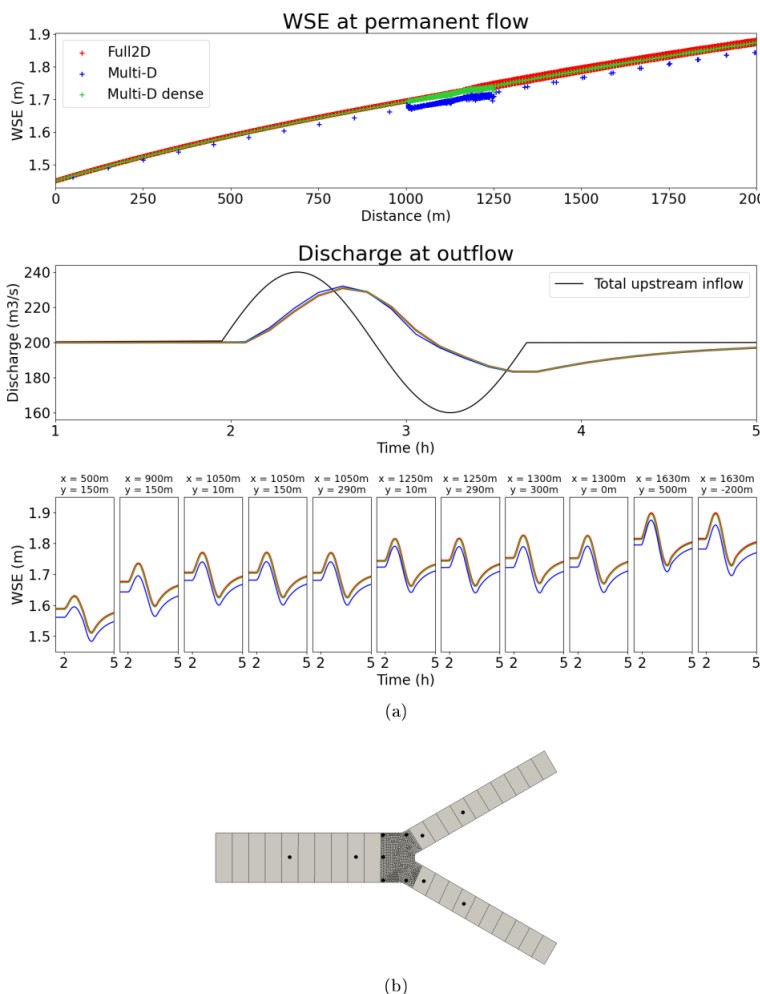

**Figure 7.** Multi-D confluence case results with second order scheme. (a) Top: Waterlines for a permanent flow of $100\,\mathrm{m}^3/\mathrm{s}$ at both upstream boundaries. In red: reference 2D waterline. In blue: waterline for the mesh (b). In green: waterline for a denser 1D2D mesh (not shown). Middle: In black: total upstream discharge during sample varied flow event (evenly distributed between upstream boundaries). In red/green/blue: downstream simulated flow for the considered meshes. Bottom: WS elevation observations for the varied flow event. (b) Multi-D mesh. Station locations are noted as black dots.



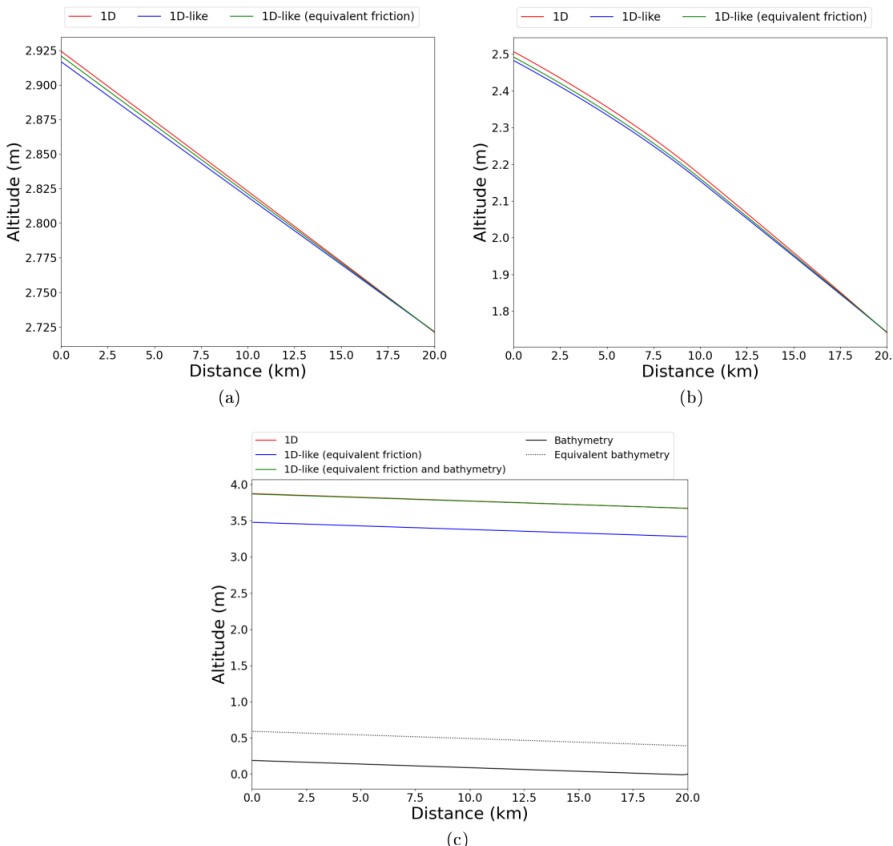

**Figure 8.** Effective friction analysis for steady waterline in academic cases: (a) rectangular channel with constant slope, (b) rectangular channel with slope break, (c) parabolic channel with constant slope. Values of effective friction parameter $\alpha$ $\left[\text{s/m}^{1/3}\right]$: (a): $5.06 \times 10^{-2}$, (b): $(5.06 \times 10^{-2}, 5.04 \times 10^{-2})$, (c): $5.46 \times 10^{-2}$. Bathymetric shift $\delta_b$ [m]: (a): 0, (b): 0, (c): 0.591. 1D reference model: fixed time step $5\,\text{s}$ spatial step $100\,\text{m}$; average Courant number equals 0.26 1Dlike models: adaptative time step with mean value of $9\,\text{s}$; 1Dlike cell length $100\,\text{m}$; average Courant number equals 0.48.

(2009). The hydraulic control generated at the slope break is well represented with a 1Dlike approach, given the aforementioned numerical errors due to the coarse grid.

Another complexification of the first case consists in changing from a rectangular XS to a parabolic one (Fig. 2). In this case,
both equivalent friction - a single homogeneous patch - and effective bathymetry, in the form of an homogeneous shift $\delta_b$ of the reference bathymetry, are needed to match the 1D WS elevation. Equivalent friction only allows to model identical wetted sections at permanent bankfull flow for a given channel width (Section 2.2.2). In this case, matching the 1D wetted section does not equate to matching its WS elevation, thus an effective bathymetry is used (Fig. 2(c)).

According to the above model comparisons, effective parameterization of channel parameters is sufficient to reproduce 1D
behaviors with a 1Dlike approach for permanent bankfull flows in simple geometries. Outside of the permanent bankfull flow, a



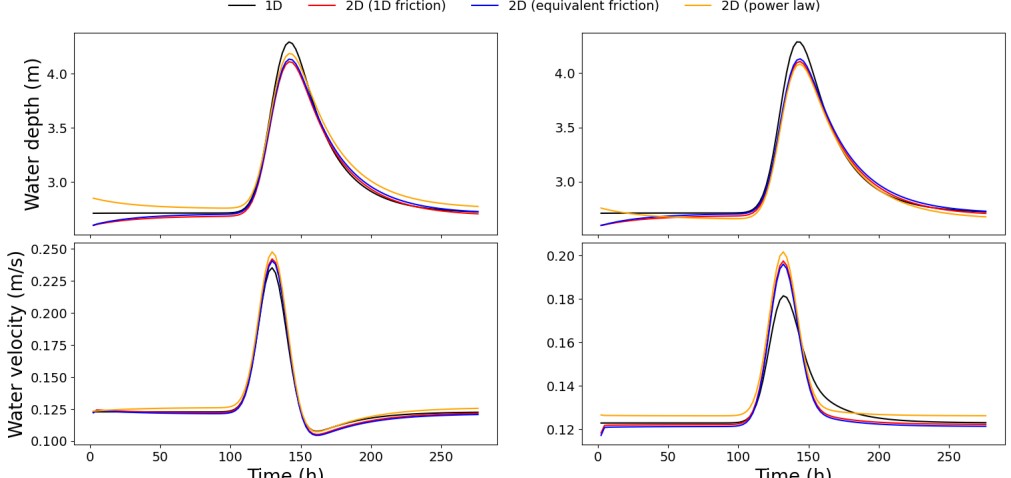

**Figure 9.** Effective friction analysis for an unsteady waterline in case (a). The power-law is given by $n = \alpha h^{\beta}$, with $\alpha = 0.0446$ and $\beta = 0.0634$. The injected hydrograph is symmetrical. Observations stations are at $x = 2.5\,\mathrm{km}$ and $x = 7.5\,\mathrm{km}$. Adaptative time step with average value of $13\,\mathrm{s}$. 1Dlike cell length: $200\,\mathrm{m}$. Average Courant number: $0.33$.

friction power-law $n = \alpha h^{\beta}$ can be used (Subsection 2.2.1). This friction aims at better 1Dlike representativity when modeled wetted surfaces (and other XS variables) are different in 1D and 1Dlike models for the same WS elevation.

**Inverse calibration**

In the following comparison, power-law parameters $\alpha$ and $\beta$ are obtained using VDA in a twin experiment setup. Observa-
tions are taken at two stations (Fig. 9). Prior values are $\alpha = 0.05\,\mathrm{s/m^{1/3}}$ and $\beta = 0$.

The classic friction law and a power-law with calibrated parameters are compared in Fig. 9 during a varied flow event. A range of water depth higher than the "bankfull" depth used to calculate equivalent friction parameters is simulated. This results in an increase in misfit to the 1D WS elevation at high flow (Fig. 9, in blue and red). The calibrated friction power-law (in orange) somewhat reduces the misfit during high flows in this simple synthetic channel and within the simulated water depth
range.

### 3.2.3  Multi-D hydraulic-hydrological data assimilation

**Temporal forcings inference**

Simple inference tests are carried on the confluence case from Subsection 3.2.1, following Pujol et al. (2020), in a twin experiment setup. A control vector $c = (Q_1(t), Q_2(t))$ is considered, where $Q_1$ and $Q_2$ are sinusoidal inflow hydrographs in-
jected at the upstream cell of the two upstream reaches. Pujol et al. (2020) shows that the minimal requirement to infer multiple spatially distributed temporal forcings simultaneously is to observe either (a) both of their unmixed signatures, at 1 point each, or (b) one of their unmixed signatures at 1 point and their mixed signatures at a second point. We verify this for configuration (b). Unmixed signatures observations are generated at a 1Dlike cell of the upstream reaches, mixed signatures observations are





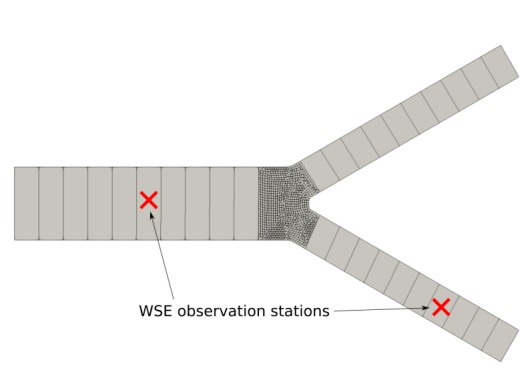

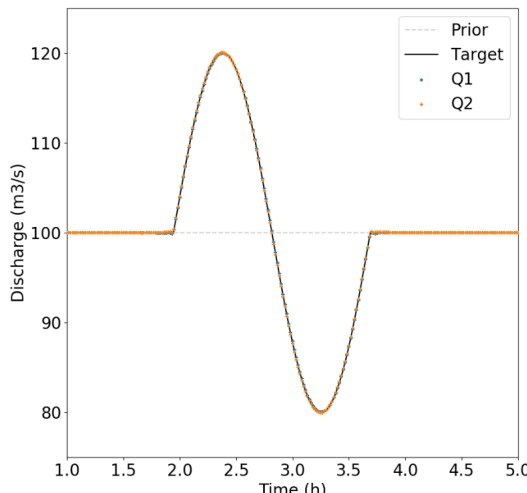

**Figure 10.** Inflow hydrographs inference on confluence synthetic case in observation setup (b) with one station on a reach upstream of the confluence and one observing mixed flows downstream in the 2D part (see mesh on the left). Given sufficient observability and an unbiased prior, inferred hydrographs (in blue and orange) are both at target (black, total overlap). The flow is subcritical with a maximum Froude value of 0.05.

generated at a cell in the 2D part (Fig. 10, left, red crosses). Prior values for the inflow hydrographs are constant and set to
their average value $\left(Q_1(t) = Q_2(t) = 100\,\mathrm{m}^3/\mathrm{s}\right)$. Using sets of two observations points, we are able to reproduce the results of Pujol et al. (2020) but with the proposed multi-D hydraulic model: the VDA algorithm enables to infer $Q_1$ and $Q_2$ close to perfectly from configurations (a) and (b) (Fig. 10, right).

**Hydrological parameters inference**

In this second twin experiment, the issue of spatially distributed calibration of a hydrological model is studied, from multi-source observations of the river network. The confluence case above is used and this time, the upstream inflows are generated by GR4H module applied on two upstream catchments inflowing the hydraulic module. For each catchment, synthetic rain and evaporation time series and a hydrological parameter set $\theta_{rr} = (c_1, ..., c_4)$ are used to generate a discharge time series over a 1-year period (not shown). Mixed observables are used: WS elevation is observed at a single downstream point and hydrological
model discharge is observed at the outlet of one of the catchments. They provide the same signal observability - mixed and unmixed signals - as in the above paragraph but with observations of different nature. Since the observed variables are of different nature and amplitude, we introduce a normalization. The cost function is here $j_{obs} = j_Z + j_Q$, with each term being normalized by the number of observations and by their range of variation such that: $j_Z = \frac{N_{o,Z}T_Z}{N_{o,Z}T_Z + N_{o,Q}T_Q}\frac{1}{(Z_{o,max} - Z_{o,min})^2}\frac{1}{N_{o,Z}}\sum\frac{1}{T_Z}\sum(Z_o(t) - Z(\theta, t))^2$ and $j_Q = \frac{N_{o,Q}T_Q}{N_{o,Z}T_Z + N_{o,Q}T_Q}\frac{1}{(Q_{o,max} - Q_{o,min})^2}\frac{1}{N_{o,Q}}\sum\frac{1}{T_Q}\sum(Q_o(t) - Q(\theta, t))^2$.



| Parameters | $c_1$ | $c_2$ | $c_3$ | $c_4$ |
|---|---|---|---|---|
| Target |  |  |  | 0.137 |
|  | 520.01 | -3.523 | 78.75 |  |
| Prior |  |  |  | 0.167 / 0.107 |
| Inferred | 520.00 / 519.46 | -3.654 / -3.820 | 80 / 80 | 0.136 / 0.137 |

**Table 2.** Hydrological parameter inference results. Prior values are taken identical to target values, except for parameter $c_4$, where under- and over-estimated prior value are considered. Inferred value for all parameters including $c_4$ are very close to the target. The "/" separates values for the 2 distinct hydrological units.

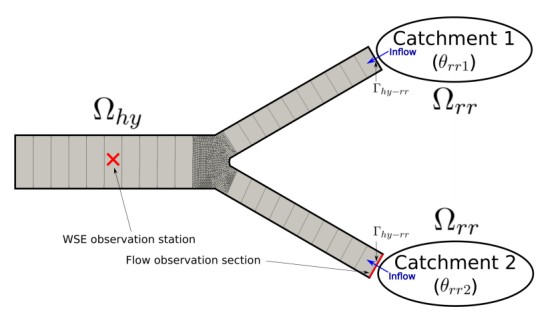
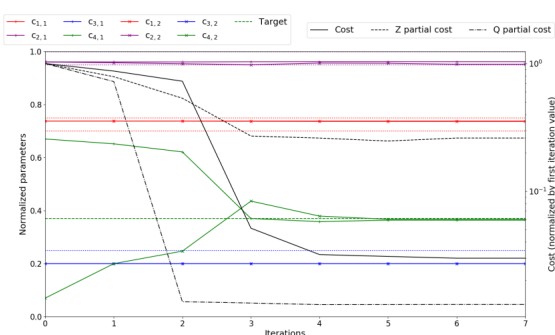

**Figure 11.** Simultaneous inference of GR4H hydrological parameters in two catchments from multi-source observations: WS elevation and flow observations. Left: Multi-D hydraulic mesh and modeled hydrological catchments. Hydrological discharge at the outlet of $\Omega_{rr}$ is inflowed upstream in $\Omega_{hy}$. Observation stations (red cross and red line) are used for infering hydrological parameters. Right: Inference of 2 sets of 4 hydrological parameters: normalized costs and parameter values over the course of the iterative optimum search. The $x_4$ background values are erroneous. $x_1$, $x_2$ and $x_4$ background values are set to target values.

Those correspond to two separate normalized squared RMSE with $N_{o,Q}$ and $N_{o,Z}$ denoting the number of observations station and $T_Q$ and $T_Z$ the number of observation time steps.

The control vector $c = (\theta_{rr1}, \theta_{rr2})$ contains the two sets of 4 hydrological parameters each. For this synthetic case, an inequality contraint of the control parameters is imposed with the bounded L-BFGS-B algorithm (Zhu et al. (1997)). Indeed, restricted research intervals are considered for the three first parameters of each catchment, namely a 5% bracket around their target values used as prior, while $c_4$ is sought in its expected variation range from an erroneous prior (Table 2). Expected ranges for GR4H parameters are provided in Le Lay (2006), for the classical GR4 formulation.

In this setup, both $j_Q$ and $j_Z$ are reduced during the iterative steps and the inferred value for both $c_4$ parameters are very close to the target value starting from an erroneous prior (Fig. 11, right). Bounded parameters $c_1$ and $c_2$ vary slightly between their bounds, while $c_3$ is locked at its lower bound from the first iteration.



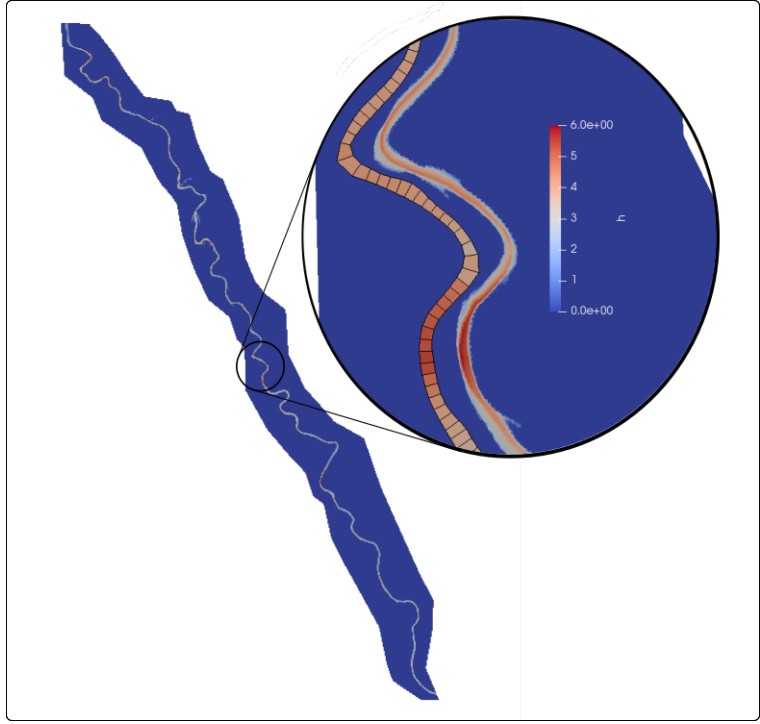

**Figure 12.** Garonne 2D model extent and simulated water depths using the 2D model and the 1Dlike model for a non-flooding event.

## 3.3 Real cases

### 3.3.1 Garonne River: 1Dlike effective model of a 2D reference case

A 1Dlike model of a reach of the Garonne river, in southern France, is built from real data and calibrated here via assimilation of spatially distributed WS observations in a twin experiment setup. A full 2D model of $75\,\mathrm{km}$ of the Garonne river (not shown, 410   $867\,500$ cells including floodplains, average edge length $\backsim 25\,\mathrm{m}$) is used to generate real-like observables on this well known case (Garambois and Monnier (2015); Brisset et al. (2018); Larnier et al. (2020); Monnier et al. (2016)). First, river extent is derived from the 2D model output at bankfull flow $Q = 400\,\mathrm{m}^3/\mathrm{s}$ with an homogeneous friction $n = 0.05\,\mathrm{s/m}^{1/3}$. This extent is in turn used to build a 1Dlike mesh, over the whole reach: single quadrangular cells cover the whole river width and are linked sequentially along the river reach (Fig. 12). 1Dlike cell interfaces are perpendicular to the flow direction, as would be 415   XSs in a 1D model. Each cell is about $100\,\mathrm{m}$ in longitudinal length. This mesh was generated using the SMS meshing tools. Cell bathymetry is first set using the lowest bathymetry point at each corresponding 2D XS.

**Permanent bankfull flow calibration**

A first expectedly imperfect 1Dlike model (Model A1) is built using the 1Dlike coarse mesh and expectedly underestimated 420   bathymetry elevation, and an homogeneous friction parameter of $0.05\,\mathrm{s/m}^{1/3}$. The simulated steady WS elevation at bankfull





flow is lower than that of the 2D model (average misfit of $0.858\,\mathrm{m}$), which is expected since the 1D bathymetry is that of the lowest point of the 2D XS (Fig. 2). Furthermore, the 1Dlike friction is underestimated and leads to an underestimated simulated water depth (and flow surface). The WS elevation misfit does not seem to follow a significant trend from upstream to downstream, although it varies sharply at points of width variation (Fig. 13, e.g. around cell 410). Recall that the goal of
this effective modeling approach is to accurately reproduce water surface signatures, including WS elevation and, tangentially, flow section (Subsection 2.2.2).

*Calibration by hand*

We here propose to reduce the misfit using, on one hand, effective friction and, on another hand, bathymetry as follows. In Model A2, the introduction of an equivalent friction parameter (Eq. 2), calculated at each 1D cell using observations from 2D
XSs, improves the WS elevation misfit (mean friction of $n = 0.062\,\mathrm{s/m^{1/3}}$, average WS elevation misfit of $0.562\,\mathrm{m}$). It reduces misfit overall, but has no significant local influence (Fig. 13). In Model A3, we use the average WS elevation misfit from Model A2 to create a simple bathymetry shift $\delta_b$ that helps fit the observations. Equivalent friction parameters from Model A2 are kept and all bathymetry points are shifted by $\delta_b = 0.562\,\mathrm{m}$ (the bottom slope is conserved). The average WS elevation misfit at bankfull flow reaches $0.003\,\mathrm{m}$ for Model A3, which is a very satisfying result.

*Calibration by VDA*

Now, the same calibration problem is addressed with an inference based on the VDA method. It is applied to the same reference model permanent flow waterline, observed at each cell. The control vector contains a single homogeneous friction parameter, as before, and spatialized bathymetry $b(x)$ for each cell. To constrain the parameter search, two VDA processes are performed separately: a bathymetry regularization and a change of variable. Inference with bathymetry regularization leads to
Model B1, with $\gamma = 1$ (Subsection 2.4). Inference with change of variable (Eq. 8) leads to Model B2, with $L_b = 500\,\mathrm{m}$ (Eq. 11). Both models lead to average misfits close to that of Model A3: $0.0839\,\mathrm{m}$ and $0.0844\,\mathrm{m}$ respectively.

Two other inference setups based on Model B1 are considered. The number of observations is divided by 10: 72 stations, homogeneously distributed at 1 per each $1\,\mathrm{km}$, are considered. In Model B1a, no regularization term is considered ($\gamma = 0$). In Model B1b, a regularization is added ($\gamma = 1$, chosen by trial and error). This weight can be optimally determined using iterative
regularization (Malou and Monnier (2022)). It is dependent on the spatial scales of observed signals and on the discretization of inferred parameters. As presented in Fig. 13 and in Table 3, both inferences lead to low misfits of the 2D permanent WS elevation. The regularization tends to reduce extreme bathymetric variations that tend to appear far from the observation points. The iterative minimization process can be followed through the cost function value and the parameter gradients (Fig. 14). Values are normalized over their initial (iteration 0) value. For Model B1a, the optimal control is reached after 40 iteration
(for a normalized cost of $8.6 \times 10^{-10}$), while for Model B1b it is reached in 85 iterations (for a normalized cost of $5.5 \times 10^{-5}$). Both models follow the same trajectory up to around iteration 12, where the regularization term $j_{reg}$ reaches the same order of scale as the WS elevation misfit term $j_{obs}$.

Variational calibration of channel parameter has allowed to fit a permanent regime water line. The following paragraphs study the reproduction of propagation in a calibrated models during a varied flow event.






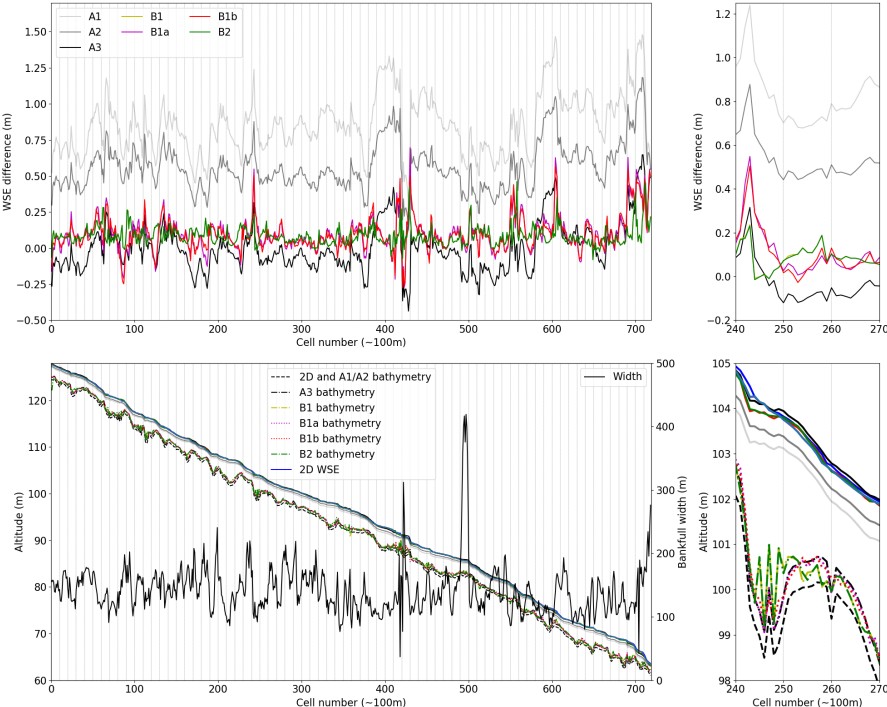

**Figure 13.** Comparison of 7 1Dlike models of $72\,\mathrm{km}$ of the Garonne river to a reference 2D SW fine model for a permanent flow (reference WS elevation in blue lines, bottom). (Up) Left: WS comparison of 1Dlike models to 2D reference model at each spatial point. Right: Zoom on $3\,\mathrm{km}$ long reach of interest. (Down) Left: WS elevation and bathymetry for the reference and 1D like models. Right: Zoom on $3\,\mathrm{km}$ long reach of interest.

For models A1, A2 and A3 (in grays), the bathymetry and homogeneous friction are manually calibrated. For model B1 (yellow), B1a (magenta), B1b (red) and B2 (green), the bathymetry and friction are calibrated by VDA. For models B1 and B2, inferences are carried out from observations at each cell (720 total). For models B1a and B1b, inferences are carried out from observations every 10 cells (i.e. around every $1\,\mathrm{km}$, 72 stations total). Vertical bars indicate these 72 stations.

**Variational calibration for a flow event**

Let us now consider a varied flow event, without flooding in the 2D model. This event last 10 days, with a peak discharge of $702\,\mathrm{m^3/s}$ at the start of day 2, a minimum discharge of $160\,\mathrm{m^3/s}$ and and average discharge of $382\,\mathrm{m^3/s}$. Given our previous inference attempts, we know that we can find a couple $(n, b(x))$ or $(n, \delta_b)$ that minimizes the average misfit. For the sake of simplicity, we consider the couple $(n, \delta_b)$. For a varied flow event, this couple would be an optimal parameterization for the average observed water line. A first inference trial is carried out for a densely observed (in space and time) varied flow. Using Model A1 as a prior for bathymetry and friction and observations from the 2D model, we find the following optimal parameters: $\left(n = 0.054\,\mathrm{s/m^{1/3}}, \delta_b = 0.669\,\mathrm{m}\right)$. The resulting optimal model is Model C.




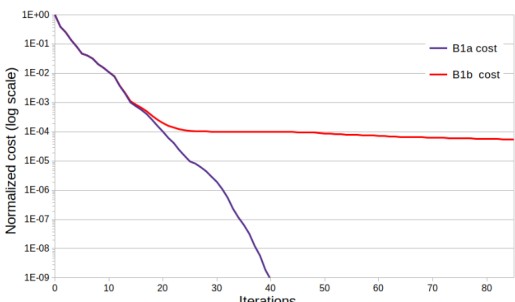
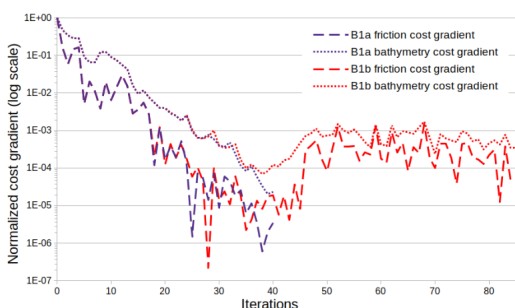

**Figure 14.** Normalized cost functions and gradients for inferences of distributed bathymetry and homogeneous friction, from 72 observation stations, in models B1a and B1b. Left: Total cost normalized by initial cost $j(\theta)/j\left(\theta^{(0)}\right)$ vs iterations. Right: Cost gradients normalized by initial gradient for the inferred parameters vs iterations.

To allow a better fit during high and low flows, we introduce the friction power-law $n = \alpha h^\beta$ (Subsection 2.2.1). Using
observations from the 2D model during the varied flow event, we infer the couple $(\beta, \delta_b)$. The value of $\alpha$ is set to $0.054\,\mathrm{s/m^{1/3}}$, the inferred value from Model D. Three different $\beta$ priors are tested: $-0.5$, $0$, $0.5$. $\delta_b$ is included in the control vector to allow modeling of more varied water depths, which may be needed to reach the optimum. All three inferences lead to the very close optimal control. Their averaging, a slight bathymetric shift $\delta_b = 0.099\,\mathrm{m}$ and $\beta = -0.017$, leads to Model D. The direct simulation results of Model D is compared to the reference model. They correspond to an average WS elevation misfit of
$0.026\,\mathrm{m}$ at high flow and of $0.11\,\mathrm{m}$ at low flow. Compared to Model C (average misfit of $0.086\,\mathrm{m}$ and $0.20\,\mathrm{m}$ at resp. high and low flow), this is an improvement. Note that NSE values for both model (Table 3) are both extremely close to $1$.

The variational calibration of the global bathymetric shift $\delta_b$ and of an homogeneous friction value $n$ in a 1D-like model has allowed to closely fit WS elevation observations of a $10$ days flood wave over the reference Garonne model. The calibration of depth-dependent friction, in the form of the $\beta$ parameter in $n = \alpha h^\beta$, has allowed an even closer fit to this reference WS
elevation over the high and low flows, i.e. a better representation of the observed flood wave propagation.

### 3.3.2 Adour basin: multi-D hydrological-hydraulic model

The whole hydrological-multi-D hydraulic tool chain is now tested in a real context both for forward and inverse problems resolution. A real and complex basin case is now considered: the Adour river basin in the South West of France, with a total drained area of $16\,890\,\mathrm{km^2}$ at the estuary. It is probably one of the most difficult basins to model in the country, because of
contrasted hydrological regimes including nival effects in the south, flash floods on small ungauged catchments, complex river network morphology, anthropized floodplains and tidal effects from downstream.

In this section, a multi-D model, composed of 1Dlike meshes and 2D zooms over floodplain area is built from available data (Fig. 15, 2D area in green). Then, forward and inverse flow simulations on the river network are presented.

**Model construction and rainfall to inundation simulation**





| Model name | Calibrated parameters | Calibration method | Flow regime | Obs. density | $n$ or $\alpha$ | $\beta$ | $\delta_b$ | Rel. misfit (PR) | RMSE (PR) | NSE (Varied) |
|---|---|---|---|---|---|---|---|---|---|---|
| 2D | - | - | - | - | 0.5 | | 0 | | | |
| A1 | - | Manual | PR | High | 0.5 | | 0 | 0.858 | 0.881 | -- |
| A2 | $n$ | | | | 0.062 | | 0 | 0.562 | 0.585 | |
| A3 | $n, \delta_b$ | | | | 0.062 | 0 | 0.562 | 0.003 | 0.167 | |
| B1 | | VDA | | | 0.059 | | | 0.084 | 0.106 | |
| B1a | $n, b(x,y)$ | | | Low | 0.059 | | - | 0.099 | 0.157 | |
| B1b | | | | | 0.057 | | | 0.099 | 0.152 | |
| B2 | | | | High | 0.059 | | | 0.084 | 0.157 | |
| C | $n, \delta_b$ | | Varied | High | 0.054 | | 0.669 | - | - | 0.99 |
| D | $\beta, \delta_b$ | | | | 0.054 | -0.017 | 0.099 | | | 0.98 |

**Table 3.** Garonne models parameters and metrics. "PR" stand for Permanent Regime. "High" observation density corresponds to 720 stations, or 1 station per 100 m. "Low" observation density corresponds to 72 stations, or 1 station per 1 km.

The following model of the Adour river combines a multi-D hydraulic network model and several hydrological models of sub-catchments (Fig. 15). It encompasses around 180 km of river reaches and includes the Adour river from its tidal boundary downstream of Bayonne up to a gauging station around 70 km upstream, and part of its main tributaries: the Nive river (around 45 km), the Oloron and Pau rivers (around 65 km in total). The river networks contains mostly single-branch reaches, with some notable flood areas around the city of Bayonne. The WS is observed in situ at 10 points, 5 of which are used as boundary conditions (Dax, Orthez, Escos, Cambo and Convergent, red points in Fig. 15). Out of the 5 remaining stations kept for data assimilation, 3 are located on river reaches (Peyrehorade, Urt and Villefranque, blue points in Fig. 15) and 2 are located in the Bayonne area (Pont-Blanc and Lesseps, blue points in Fig. 15).

At the 4 upstream points of the modeled river network, 4 sub-catchments are modeled with 4 lumped hydrological models (Subsection 2.3). Their drainage areas are respectively about 7811, 842, 2480 and 2464 km$^2$. The hourly discharge have been extracted from the HYDRO[2] database while the rainfall data from the radar observation reanalysis ANTILOPE J+1, which merges radar and in situ gauge observations, is provided by Météo France. The interannual temperature data is provided by the SAFRAN reanalysis, Météo France, and then used to calculate the potential evapotranspiration using the Oudin formula Oudin et al. (2005). The rainfall and PET are at a spatial resolution of 1 km$^2$ square grid, and processed into hourly time step. Spatial averages of the rainfall and PET computed with SMASH distributed platform (Jay-Allemand et al. (2020); Colleoni et al. (2021)) over the 4 catchments are used as inputs for the lumped GR4H model. Lumped parameter sets for the 4 GR4H models

---

[2]http://www.hydro.eaufrance.fr ; french ministry in charge of environment



are simply obtained here using for each the airGR global calibration algorithm (Coron et al. (2017)). In this section, inferences are carried out only for upstream inflow hydrographs, not hydrological parameters. Indeed, the study of global calibration and regionalization issues of spatially distributed hydrological models is left for further research.

Our multi-D hydraulic modeling approach (Section 2) is applied to this complex case as follows. First, a "1Dlike-only" model of the whole network is built. Then, a multi-D model is built based on the "1Dlike-only" model, with the addition of a 2D mesh of a floodplain (Fig. 16, left).

On the "1Dlike-only" model, the goal is to analyze 1Dlike signal propagation representation at a low computational cost. The Adour 1Dlike model is built similarly to the Garonne 1Dlike model, using DEM data to determine minor bed bank line

placement and build the quadrangular cells. Bathymetry comes from $25 \times 25\,\mathrm{m}$ DEM data, aggregated from fine LiDAR data from public databases, extracted at each cell. This rough approximation is sufficient to show the potential of the DassFlow assimilation tool chain on a large scale river network. This "1Dlike-only" model contains 1409 cells.

Then, a multi-D model is obtained: the existing 2D mesh from a Telemac model is coupled to a 1Dlike mesh, similarly to the reference Telemac-Mascaret model from the regional flood forecast center SPC-GAD. The 1Dlike parts of the mesh are

kept identical to the "1Dlike-only" model, while the 2D part is the mesh extracted from the Telemac-Mascaret model (Fig. 16, left) provided by SPC-GAD. For hydraulic coherence, bathymetry at coupled 1Dlike cells are taken as the average bathymetry of the linked 2D cells. This "1D2D" model contains $66\,982$ cells in the 2D area and 1342 cells in 1Dlike reaches. Results for a flooding event in the Bayonne 2D area are presented in Fig. 16, on the right. Modeled variables appear coherent over the 1D2D area and a high resolution floodmap in coherence with flow conditions in the whole river network is obtained.

A 1 day (physical time) flooding event is computed in around $20\,\mathrm{min}$ (computation time) in the "1D2D" model (with $68\,324$ total cells, 6 threads in parallel). The same event leads to a $7\,\mathrm{s}$ computation time for the "1Dlike-only" model (with 1409 cells, same computational setup). Remember that computation time may vary depending on the number of wetted cells and on the adaptative time step calculation. This relatively low computation time, and the potential to decrease it further by using more threads in parallel, indicate that this multi-D method is suited to operational use. The code version this work is based on was

proven scalable (Couderc et al. (2013)). Additions made to the current version should not change this, but no numerical testing has been done.

**Assimilation of WS observables to infer 4 upstream hydrographs**

To investigate the 1Dlike effective modeling on a river network, a twin experiment setup is designed to infer large control

vector from a realistic observability on the "1Dlike-only" model.

The considered control vector is composed of the 4 upstream hydrographs $c = (Q_{Dax}(t), Q_{Esc}(t), Q_{Ort}(t), Q_{Cam}(t))$. Observations of WS elevation are generated at Peyrehorade, Urt and Villefranque stations, and additionally at a virtual station directly downstream from Orthez. These 4 points give theoretically sufficient observability to identify the 4 upstream hydrographs. Indeed, they sample mixed and unmixed signal similarly to the academic setup in Subsection 3.2.1). The observation

pattern also corresponds to a reasonable expectation of spatial observability in French river networks. Prior hydrograph values are classically set to a constant average discharge value.



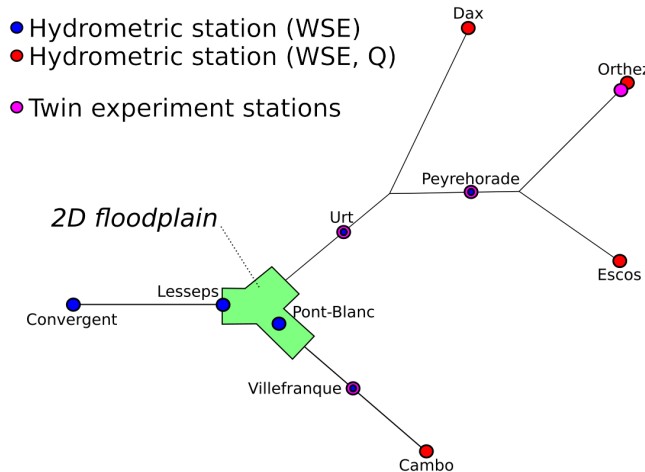

**Figure 15.** Schematic view of the complete Adour river network and observability. Dimensions are not respected: total river lengths equal $\approx 180\,\text{km}$, 2D floodplain area equals $\approx 5 \times 3\,\text{km}^2$. Tidal BC influence (from the downstream BC at Convergent) is observed up to Dax (and further upstream), Peyrehorade and Villefranque.

Simultaneous inference of the 4 hydrographs is satisfying. As shown in Fig. 17, left, upstream hydrographs injected at Cambo and Orthez are inferred very accurately ($NSE > 0.95$). This is due to their WS elevation signature being observed "unmixed" in the respective downstream reaches. The Escos hydrograph WS signature is observed at Peyrehorade, mixed with
the Orthez hydrograph WS elevation signature. This leads to partial error of signature attribution and a less accurate inference ($NSE = 0.57$). The Dax hydrograph WS elevation signature is observed only at the Urt station, where its signal is mixed with that of Escos and Orthez. The resulting inference is the less accurate with a $NSE$ of $0.50$ and closer in shape to the inferred Escos and Orthez hydrographs. This hints at correlated influences of these hydrographs at observation stations and insufficient observability of the Dax hydrograph signal given the model hydrodynamics. Observed signals at the 4 stations are however all
very accurate (Fig. 17, right). For upstream stations (Orthez and Villefranque), this is due to the accurate inference of upstream hydrographs. For stations under the influence of the tidal BC (Peyrehorade and Urt), this is due to the backwater influence of the BC, which compensates for the error in inferred hydrographs (namely at Dax, and less so at Escos).

In conclusion, this experiment shows the capability of the VDA tool chain to infer various upstream hydrographs on 1Dlike network models. Note that multi-D models are identical to the investigated 1Dlike model in terms of VDA capabilities. It paves
the way towards investigations on multi-variate inferability of spatially distributed hydraulic-hydrological parameters.

In the future, investigation on the influence of data sparsity, observation weights and ill-posed problem constraints should be carried out.



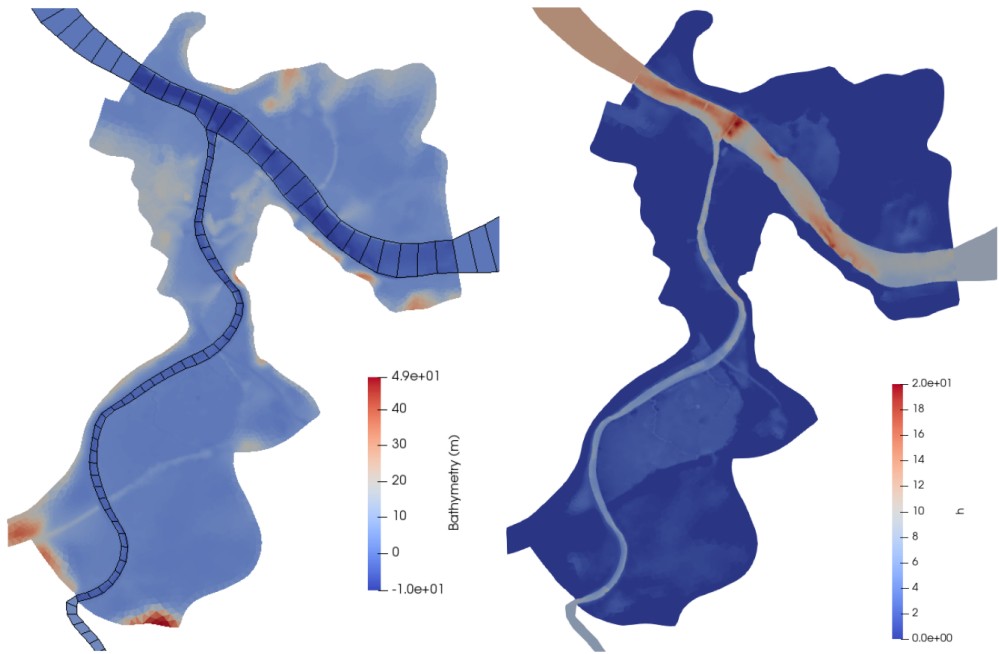

**Figure 16.** Zoom on the Bayonne area (green area in Fig. 15) using the 1Dlike and 1D2D approaches. Manning values are homogeneous throughout the network and floodplains at $0.05\,s/\mathrm{m}^{1/3}$. Left: Bathymetry in 1Dlike and 2D meshes. The 2D area has $66\,982$ cells and the coupled 1Dlike reaches account for 1342 cells. The south, east and west interfaces respectively feature 6, 18 and 14 2D cells. The exclusively 1Dlike mesh contains 1409 cells. Right: Simulated water height for the 1D2D model for a sample flooding event (low tide, $Q_{Adour} = 650\,\mathrm{m}^3/\mathrm{s}$ and $Q_{Nive} = 58\,\mathrm{m}^3/\mathrm{s}$ at 1D2D interfaces).

## 4   Conclusions and perspectives

This article presents a new approach and numerical chain for the multi-D hydrological-hydraulic modeling of complex river
networks with variational data assimilation capabilities. It is based on the VDA algorithm and the finite volume solvers (including second order one and accurate treatment of wet/dry front propagation) from Monnier et al. (2016). The resolution of the full 2D shallow water equations (Eq. 1) is performed with a single finite volume solver applied on a multi-D discretization of a river network domain. This lattice consists in "1Dlike" reaches meshed with irregular quadrangular cells connected, via 1Dlike-2D interfaces, to 2D zooms consisting in higher resolution unstructured meshes - either triangular or quadrangular. This
hydraulic model is inflowed with a hydrological model enabling to describe upstream/lateral catchments inflow hydrographs. In this work, the parsimonious GR4 model is integrated (Perrin et al. (2003)), in its state space version (Santos et al. (2018)) for the sake of differentiability (for the VDA computations). This approach is implemented in the platform DassFlow (Monnier et al. (2019)). The adjoint of the whole tool chain is generated with Tapenade (Hascoet and Pascual (2013)) and validated. Forward-inverse capabilities with the new components is assessed on several cases of increasing complexity.

 

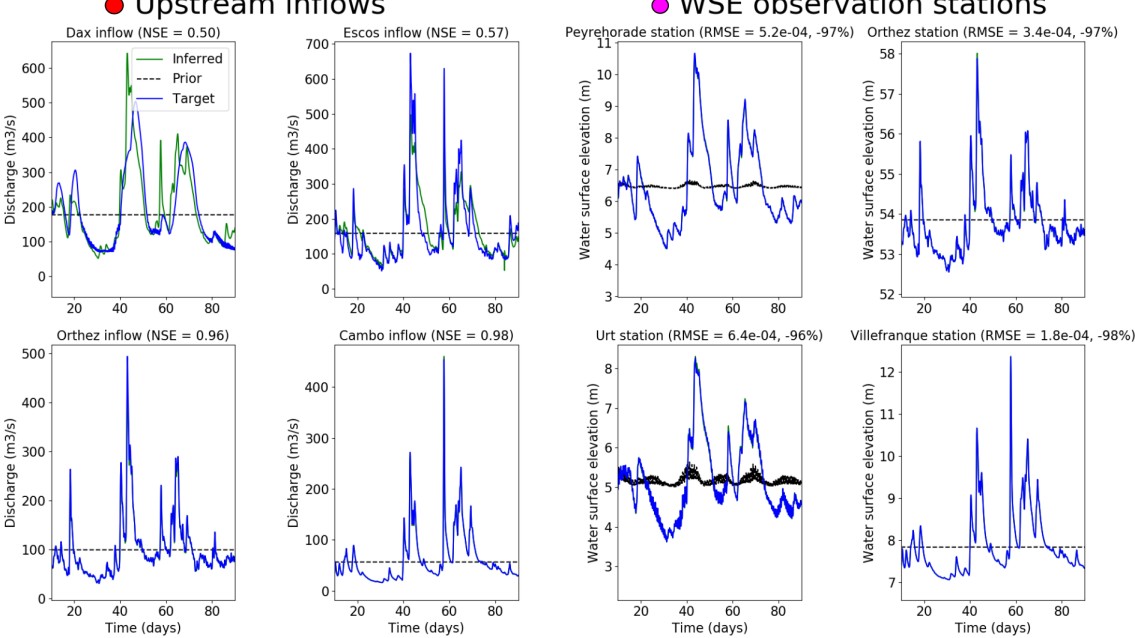

**Figure 17.** Upstream hydrograph inferences from 4 observation stations in twin experiment setup. Left: In blue: upstream inflow hydrographs used to generate observations. In green: inferred hydrographs at 20 iterations. Dotted lines: background hydrograph value given for iteration 0 (mean target inflow over assimilation window). NSE values are given for their informative value but are not used in the assimilation process. Right: In blue: target WS elevation. In green: WS elevation generated by the inferred hydrographs. Dotted lines: WS elevation generated by the background hydrographs. Downstream tidal boundary influence is felt at the two most downstream stations (Urt and Peyrehorade). RMSE value given are used to calculate cost in the assimilation process (Section 2.4).

The 1Dlike effective hydraulic modeling approach as well as its coupling with higher resolution 2D zooms was validated, against: 1) reference 1D and 2D hydraulic models, on an array of academic; 2) complex real cases featuring 1D and 2D flow variabilities in river networks with confluences and floodplains. Those cases are also employed to test the coupling with the hydrological model implemented in a semi-distributed setup.

     Considering single (resp. multiple) type(s) of flow signatures observations in those river networks through a single-(resp. 570 multi)-objective observation cost function, the capabilities of the VDA method for infering mono/multi-variate control vectors of large dimensions was successfully tested.

     From the obtained results, the following conclusions can be raised:

1. A complete integrated multi-D model coupled with an hydrological model has been implemented and validated in a parallel environment. Moreover, this complete tool chain includes VDA capabilities based on the adjoint code.



2. The 1Dlike modeling approach enables to simulate fine physical flow states compared to reference 1D or 2D SW models; hydrograph propagation remains very close also.

3. The inference, from heterogeneous observations in the river network, of multi-variate controls among multiple inflow discharge hydrographs, bathymetry, friction of the multi-D hydraulic model but also hydrological model parameters is demonstrated. Very accurate inferences are obtained when the available information contained in system observability and priors is sufficient regarding the nature and quantity of unknown parameters (see discussion in Brisset et al. (2018); Larnier et al. (2020); Garambois et al. (2020) and references therein).

4. Information feedback from the river network to upstream hydrological models of sub-catchments is shown.

5. Real flows on complex channel geometries can be accurately simulated with the 1Dlike model, despite its intrinsic rectangular XS, thanks to the calibration of effective geometry-friction patterns. The depth-dependent friction law helps to reduce misfits across flow regimes.

6. High resolution simulations of real flows can be obtained on complex river networks including floodplains and confluences with reduced simulation costs.

To our knowledge, the present numerical tool is the first one proposing, large scale multi-D river network modeling with VDA capabilities.

Short term perspectives will aim to taylor the data assimilation algorithm to perform complex data assimilation experiments at basin scale using various multi-source datasets. To be actually operational, improvements pertaining to the construction of 1Dlike models from global public databases are needed to deploy the multi-D approach to a large number of river networks. Coupling is ongoing with the SMASH spatially distributed hydrological platform (Jay-Allemand et al. (2020); Colleoni et al. (2021)) on which is based the French flash flood warning system VigiCrues Flash (Garandeau et al. (2018)). Further work will also test the integrated chain on flash flood Mediterranean basins as well as on larger basins in a satellite observability context. The implementation of porosity models (Guinot et al. (2018) and references therein) represents a very interesting research direction regarding effective floodplains and 1Dlike reaches parameterizations. This is especially true with depth-dependent porosity (Özgen et al. (2017) and references therein) applied to complex channel geometries and with spatially distributed calibration.

## Appendix A: 2D $(h, u, v)$ Shallow Water scheme

### A1   2D solver

Recall the rotational invariance property of the SWE (Eq. 1) which simplifies the sum of 2D problems in Eq. (3) to 1D Riemann problems. The fluxes $\mathbf{F}_e$ are computed using a Riemann solver. Each local Riemann problem depends on left and right states at the interface $e$.





The HLLC approximate Riemann solver is used. This gives the expressions:

$$
\begin{cases}
\left[\hat{\mathbf{F}}_e^{HLLC}\right]_{1,2} = \dfrac{s_{K_e}\left[\mathbf{F}\left(\hat{\mathbf{U}}_K\right)\right]_{1,2} - s_K\left[\mathbf{F}\left(\hat{\mathbf{U}}_{K_e}\right)\right]_{1,2} + s_K s_{K_e}\left(\left[\hat{\mathbf{U}}_{K_e}\right]_{1,2} - \left[\hat{\mathbf{U}}_K\right]_{1,2}\right)}{s_{K_e} - s_K} \\[2ex]
\left[\hat{\mathbf{F}}_e^{HLLC}\right]_3 = \left[\hat{\mathbf{F}}_e^{HLLC}\right]_1 \hat{v}* \quad \text{with} \quad \hat{v}* = \begin{cases} \hat{v}_K & \text{if} \quad s^* \geq 0 \\ \hat{v}_{K_e} & \text{if} \quad s^* < 0 \end{cases}
\end{cases}
\tag{A1}
$$

where the wave speed expression is those proposed in Vila (1986):

$$
\begin{aligned}
s_K &= \min\left(0, \hat{u}_K - \sqrt{gh_K}, \hat{u}_{K_e} - 2\sqrt{gh_{K_e}} + \sqrt{gh_K}\right) \\
s_{K_e} &= \max\left(0, \hat{u}_{K_e} + \sqrt{gh_{K_e}}, \hat{u}_K + 2\sqrt{gh_K} - \sqrt{gh_{K_e}}\right)
\end{aligned}
\tag{A2}
$$

It has been demonstrated that this insures $L^\infty$ stability, positivity and consistency with entropy condition under a CFL
condition.

For the intermediate wave speed estimate, following Toro (2001) we set:

$$
s^* = \frac{s_K h_{K_e} \hat{u}_{K_e} - s_{K_e} h_K \hat{u}_K - s_K s_{K_e}\left(h_{K_e} - h_K\right)}{h_{K_e}\left(\hat{u}_{K_e} - s_{K_e}\right) - h_K\left(\hat{u}_K - s_K\right)}
\tag{A3}
$$

A Courant–Friedrichs–Levy (CFL) condition for the time step $\Delta t^n$ applies, see e.g. Vila and Villedieu (2003).

### A1.1 Well-balancing

The numerical scheme must preserve the fluid at rest property, that is the gradient of bathymetry $\nabla z_b$ must not provide $\mathbf{u}^{n+1} \neq$
0 if $\mathbf{u}^n = 0$. In the presence of topography gradients (in particular those perpendicular to the streamlines) the basic topography
gradient $\nabla z_b$ discretization in the gravity source term $\mathbf{S}_g(\mathbf{U})$ generates spurious velocities. There is no discrete balance between
the hydrostatic pressure and the gravity source term anymore: $\nabla\left(gh^2/2\right) \neq -gh\nabla z_b$.

Following Audusse et al. (2004), this issue is solved by considering the following change of variable.

From now, we consider the water depth $h_{e,K}^*$ defined from the "reconstructed" topography $z_e$ (Fig. A1) at edge $e$ as:

$$
\begin{aligned}
&h_{e,K}^* = \max\left(0, h_{e,K} + z_{e,K} - z_e\right) \\
&\text{with} \quad \begin{cases} z_{e,K} = \eta_{e,K} - h_{e,K} \\ z_e = \max\left(z_{e,K}, z_{e,K_e}\right) \end{cases}
\end{aligned}
\tag{A4}
$$

The conservative variable vector $\mathbf{U}_{e,K}^n$ in the semi-discrete scheme (Eq. 3) is now considered with the new variable:

$$
\mathbf{U}_{e,K}^* = \begin{bmatrix} h_{e,K}^* \\ h_{e,K}^* \mathbf{u} \end{bmatrix}
\tag{A5}
$$





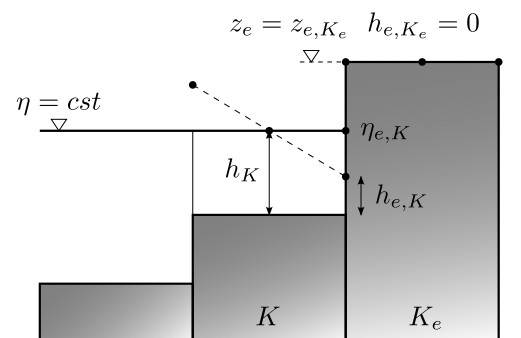

**Figure A1.** Typical situation of desired well-balanced property in presence of a wet/dry front. The WS elevation is here denoted by $\eta$ (from REFDOC).

Note that this new variable $h_{e,K}^*$ depends on the bathymetry values $(z_{e,K}, z_e)$.

The resulting well-balanced first order scheme reads:

$$\mathbf{U}_K^{n+1} = \mathbf{U}_K^n - \frac{\Delta t^n}{m_K} \sum_{e \in \partial K} m_e \left( \mathbf{F}_e(\mathbf{U}_{e,K}^{*n}, \mathbf{U}_{e,K_e}^{*n}, \mathbf{n}_{e,K}) + \mathbf{S}_p \left( \mathbf{U}_{e,K}^n, \mathbf{U}_{e,K}^{*n}, \mathbf{n}_{e,K} \right) \right) \tag{A6}$$

with

$$\mathbf{S}_p \left( \mathbf{U}_{e,K}^n, \mathbf{U}_{e,K}^{*n}, z_K, z_{e,K}, \mathbf{n}_{e,K} \right) = \begin{bmatrix} 0 \\ \frac{g}{2} \left( \left( h_{e,K}^{n*} \right)^2 - \left( h_{e,K}^n \right)^2 \right) \mathbf{n}_{e,K} \end{bmatrix} \tag{A7}$$

### A1.2    Prediction-correction time scheme

The friction source term is taken into account in the complete SW system by deriving a prediction-correction time scheme, see e.g. Toro (2001).

We denote here by $\bar{\mathbf{U}}_K^{n+1}$ the FV solution at time $t^{n+1}$ of the (well-balanced) scheme, either first or second order, of the SW system with the gravitational term $\mathbf{S}_g$ but without the friction term $\mathbf{S}_f$. Recall that we denote: $\mathbf{U} = (h, h\boldsymbol{u})^T$.

At each time step, from $n$ to $n+1$,

– **Step 1:** computation of $\bar{\mathbf{U}}^{n+1}$, solution of the conservative SW system, i.e. the SW system *without* $S_f$, i.e. the FV solution of the following system:

$$\partial_t \boldsymbol{U} + \partial_x \boldsymbol{F}(\boldsymbol{U}) + \partial_y \boldsymbol{G}(\boldsymbol{U}) = \mathbf{S}_g(\boldsymbol{U}) \tag{A8}$$

– **Step 2:** given the "predicted value" $\bar{\mathbf{U}}^{n+1}$, compute $\mathbf{U}^{n+1}$ solution of:

$$\partial_t \mathbf{U} = \mathbf{S}_f(\mathbf{U}) \tag{A9}$$



General schemes (explicit, implicit or semi-implicit ones) including the friction source term $\mathbf{S}_f$ in the discretization of the model (Eq. 1) can be written as: for all $K$,

$$\mathbf{U}_K^{n+1} = \bar{\mathbf{U}}_K^{n+1} + \Delta_t^n \mathbf{S}_f \left( \bar{\mathbf{U}}_K^{n+1}, \bar{\mathbf{U}}_{K_e}^{n+1} \right) \tag{A10}$$

Note that this splitting scheme is consistent at first order in $\Delta t$ with the complete SWE. Splitting scheme second order in time is possible; it is not detailed later.

In the case of the Manning-Strickler law, the friction term reads $\mathbf{S}_f = -gn^2 \begin{bmatrix} 0 \\ \frac{|\bar{\boldsymbol{u}}|}{h^{\frac{1}{3}}} \bar{\boldsymbol{u}} \end{bmatrix}$.

Therefore the equation to be solved (Eq. A9) reads:

$$\partial_t \begin{pmatrix} h \\ h\bar{\boldsymbol{u}} \end{pmatrix} = -gn^2 \begin{pmatrix} 0 \\ \frac{|\bar{\boldsymbol{u}}|}{h^{\frac{1}{3}}} \bar{\boldsymbol{u}} \end{pmatrix} \tag{A11}$$

Since the friction source term $\mathbf{S}_f$ is zero in the mass conservation equation, we remark that $h^{n+1} = \bar{h}^{n+1}$. As a consequence, we consider the non-zero momentum component only $\partial_t (h\bar{\boldsymbol{u}}) = -gn^2 \frac{|\bar{\boldsymbol{u}}|}{h^{\frac{1}{3}}} \bar{\boldsymbol{u}}$

### A1.3 First order expression of $\mathbf{U}^{n+1}$

Let us consider the implicit scheme :

$$\frac{h^{n+1}\boldsymbol{u}^{n+1} - h^{n+1}\bar{\boldsymbol{u}}^{n+1}}{\Delta t^n} = -gn^2 \frac{|\boldsymbol{u}^{n+1}| \boldsymbol{u}^{n+1}}{(h^{n+1})^{\frac{1}{3}}} \tag{A12}$$

This implies that:

$$|\boldsymbol{u}^{n+1}| \boldsymbol{u}^{n+1} + \frac{(h^{n+1})^{\frac{4}{3}}}{\Delta t^n gn^2} \left( \boldsymbol{u}^{n+1} - \bar{\boldsymbol{u}}^{n+1} \right) = 0 \tag{A13}$$

Let us set $c = \frac{(h^{n+1})^{\frac{4}{3}}}{\Delta t^n gn^2}$, $c \geq 0$. Note that $|\boldsymbol{u}^{n+1}| \boldsymbol{u}^{n+1} + c\boldsymbol{u}^{n+1} = c\bar{\boldsymbol{u}}^{n+1}$. Therefore for non vanishing velocities, it exists $\alpha \in$ $]0,1]$ such that: $\boldsymbol{u}^{n+1} = \alpha\bar{\boldsymbol{u}}^{n+1}$. Adopting these notations, we obtain $|\bar{\boldsymbol{u}}^{n+1}| \bar{\boldsymbol{u}}^{n+1} \alpha^2 + c\bar{\boldsymbol{u}}^{n+1}\alpha - c\bar{\boldsymbol{u}}^{n+1} = 0$. This simplifies to:

$$|\bar{\boldsymbol{u}}^{n+1}| \alpha^2 + c\alpha - c = 0 \tag{A14}$$

Since $\alpha \geq 0$, the root of this quadratic equation reads:

$$\alpha = \frac{-c + \sqrt{c^2 + 4c|\bar{\boldsymbol{u}}^{n+1}|}}{2|\bar{\boldsymbol{u}}^{n+1}|} \tag{A15}$$




Let us define the function $\varepsilon = \frac{1}{c}\left|\bar{\mathbf{u}}^{n+1}\right| = \Delta t^n g n^2 \frac{\left|\bar{\mathbf{u}}^{n+1}\right|}{(h^{n+1})^{4/3}}$. Observe that $\varepsilon = O\left(\Delta t^n\right)$, also $\varepsilon = O\left(\frac{\left|\bar{\mathbf{u}}^{n+1}\right|}{(h^{n+1})^{4/3}}\right)$. By adopting this notation, Eq. (A15) reads: $\alpha = \frac{\sqrt{1+4\epsilon}-1}{2\epsilon}$. After some rearrangements, we obtain $\alpha = \frac{2}{1+\sqrt{1+4\varepsilon}}$. At first order in $\epsilon$, we get $\alpha \sim \left(\frac{1}{1+\epsilon/4}\right) \sim 1 - \epsilon/4$. Finally, we obtain:

$$\mathbf{U}_K^{n+1} = \begin{pmatrix} h^{n+1} \\ h^{n+1}\boldsymbol{u}^{n+1} \end{pmatrix} = \begin{bmatrix} \bar{h}^{n+1} \\ h^{n+1}\bar{\mathbf{u}}^{n+1}\left( \frac{2\left(\bar{h}^{n+1}\right)^{2/3}}{\left(\bar{h}^{n+1}\right)^{2/3} + \sqrt{\left(\bar{h}^{n+1}\right)^{\frac{4}{3}} + 4\Delta t^n g n^2 \left|\bar{\mathbf{u}}^{n+1}\right|}} \right) \end{bmatrix} \tag{A16}$$

with $\left|\bar{\mathbf{u}}^{n+1}\right|$ the solution of Eq. (A8).

### A1.4   Second order scheme

In order to obtain a globally second order scheme, a higher-order time stepping scheme is needed. Let us briefly describe the ingredients of this second order well-balanced positive scheme that is strictly the same as the one proposed in Couderc et al. (2013); Monnier et al. (2016). Actual second order accuracy, considering source terms $\mathbf{S}_g$ and $\mathbf{S}_f$, is achieved through the
combination of a Monotonic Upwind Scheme for Conservation Laws (MUSCL) spatial reconstruction and an IMEX RK time scheme (see Monnier et al. (2016) and references therein), as well as a spatial discretization of $\mathbf{S}_g$ and the semi-implicitation friction source term $\mathbf{S}_f$ given by in the subsection above.

A monoslope second order MUSCL scheme is adopted, see e.g. Chévrier and Galley (1993); Buffard and Clain (2010). It leads to new expressions of $\mathbf{U}_K^n$ and $\mathbf{U}_{K_e}^n$. With this linear reconstruction, one can expect a scheme with a second-order
accuracy in space (for regular solutions only). In order to prevent large numerical dispersive instabilities, the computed vectorial slopes are limited by applying a maximum principle. Furthermore, to handle the presence of wet-dry fronts that can break the Finite Volume mass conservation property, a Barth limiter (Barth (2003)) is employed.

### Appendix B:  1D $(A, Q)$ Saint-Venant

A river network $\Omega^{1D}$ is described by connected closed line segments. For a flow XS orthogonal to the main (longitudinal) flow
direction of curvilinear abscissa $x \in \Omega^{1D}$ (distance from downstream), at time $t \in [0, T]$, let $A(x,t)$ be the flow cross-sectional area $\left[\mathrm{m}^2\right]$ and $Q(x,t)$ the discharge $\left[\mathrm{m}^3/\mathrm{s}\right]$ such that $Q = UA$ with $U(x,t)$ defined as the longitudinal XS averaged velocity $[\mathrm{m/s}]$. The 1D Saint-Venant equations in $(A, Q)$ variables at a flow XS write as follows:

$$\begin{cases} \frac{\partial A}{\partial t} + \frac{\partial Q}{\partial x} = 0 \\ \frac{\partial Q}{\partial t} + \frac{\partial}{\partial x}\left(\frac{Q^2}{A}\right) + gA\frac{\partial Z}{\partial x} = -gA\frac{|Q|Q}{K^2 A^2 R_h^{4/3}} \end{cases} \tag{B1}$$

where $Z(x,t)$ is the WS elevation $[m]$ and $Z = (b + h)$ with $b(x)$ $[m]$ the river bed level and $h(x,t)$ $[m]$ the water depth ,
$R_h(x,t) = A/P_h$ $[m]$ the hydraulic radius , $P_h(x,t)$ $[m]$ the wetted perimeter, $g$ $\left[m/s^2\right]$ is the gravity magnitude. Let us recall



the Froude number definition $\mathrm{Fr} = U/c$ comparing the average flow velocity $U$ to pressure wave celerity $c = \sqrt{\frac{gA}{W}}$ where $W$ $[m]$ is the flow top width.

The friction term $S_f$ is classically parameterized with the empirical Manning-Strickler law established for uniform flows $\dfrac{|Q|\,Q}{K^2 A^2 R_h^{4/3}}$ where $K\ \left[\mathrm{m}^{1/3}/\mathrm{s}\right]$ is the Strickler coefficient.

The Saint-Venant equations are solved on each segment of the river network and the continuity of the flow between segments is ensured by applying an equality constrain on water levels at the confluence between two segments.

Boundary conditions are classically imposed (subcritical flows here) at boundary nodes with inflow discharges $Q(t)$ at upstream nodes and WS elevation $Z(t)$ at the downstream node; lateral hydrographs $q_l(t)$ at in/outflow nodes. The initial condition is set as the steady state backwater curve profile $Z_0(x) = Z(Q_{in}(t_0), q_{l,1..L}(t_0))$ for hot-start. This 1D Saint-Venant

model is discretized using the classical implicit Preissmann scheme (see e.g. Cunge et al. (1980)) on a regular grid of spacing $\Delta x$ using a double sweep method enabling to deal with flow regimes changes with a one-hour time step $\Delta t$. This is implemented into the computational software DassFlow (Brisset et al. (2018); Larnier (2010)).

The numerical scheme is a semi-implicit finite difference scheme (generalized Preissmann scheme) with a double sweep Local Partial Inertial method to minimize the inertial terms (see documentation in Brisset et al. (2018); Larnier (2010)).




## Appendix C: GR4 hydrological model operators

The state-space version of the lumped conceptual hydrological model GR4 presented in Santos et al. (2018) consists in the set of ordinary differential equations (ODE) given in Eq. (4) and recalled here for clarity:

$$\frac{d\boldsymbol{h}}{dt} = \begin{cases} \dot{h}_p & = P_s - E_s - P_{erc} \\ \dot{h}_1 & = P_r - Q_{Sh,1} \\ \dot{h}_2 & = Q_{Sh,1} - Q_{Sh,2} \\ ... & ... \\ \dot{h}_{nres} & = Q_{Sh,nres-1} - Q_{Sh,nres} \\ \dot{h}_r & = Q_9 + F - Q_r \end{cases}$$

They involve the following fluxes:

$$\begin{cases} E_s = & E\left(\frac{2h_S}{c_1} - \left(\frac{h_p}{c_1}\right)^\alpha\right) \\ P_S = & P\left(1 - \left(\frac{h_p}{c_1}\right)^\alpha\right) \\ P_{erc} = & \left(\frac{\nu}{c_1}\right)^{\beta-1} \frac{1}{\beta-1}\left(h_p^+\right)^\beta \\ P_R = & P\left(\frac{h_p}{c_1}\right)^\alpha + \left(\frac{\nu}{c_1}\right)^{\beta-1}\frac{1}{\beta-1}\left(h_p^+\right)^\beta \\ Q_{N1} = & \frac{n_{res}-1}{c_4}h_1^+ \\ ... & ... \\ Q_{N11} = & \frac{n_{res}-1}{c_4}h_{n_{res}}^+ \\ Q_9 = & \Phi\frac{n_{res}-1}{c_4}h_{n_{res}}^+ \\ Q_1 = & (1-\Phi)\frac{n_{res}-1}{c_4}h_{n_{res}}^+ \\ F = & \frac{c_2}{c_3^\omega}\left(h_r^+\right)^\omega \\ Q_D = & (1-\Phi)\frac{n_{res}-1}{c_4}h_{n_{res}+1} - \frac{c_2}{c_3^\omega}\left(h_r^+\right)^\omega \\ Q_R = & \frac{1}{\left(h_R^+\right)^{\gamma-1}(\gamma-1)}\left(h_r^+\right)^\gamma \end{cases}$$

The following parameter are set following Perrin et al. (2003) and Santos et al. (2018): $\alpha = 2$, $\beta = 5$, $\gamma = 5$, $\omega = 3.5$, $\nu = 4/9$, $\Phi = 0.9$, $n_{res} = 11$.

Calibrated parameter for the Adour case were obtained using the airGR global calibration algorithm (Coron et al. (2017)) from the freely available package[3].

---

[3]https://webgr.inrae.fr/logiciels/airgr/





| Parameter | Adour | Oloron | Pau | Nive |
|-----------|-------|--------|-----|------|
| $x_1$ | 413.744 | 1844.567 | 1118.78 | 982.401 |
| $x_2$ | 0.148 | 1.363 | 1.134 | 0.696 |
| $x_3$ | 86.418 | 117.919 | 112.168 | 90.017 |
| $x_4$* | 55.466 | 12.739 | 11.059 | 6.980 |

**Table C1.** Calibrated hydrological parameters of the 4 upstream hydrological catchments from the Adour multi-D hydrographic network model. *Note that the $x_4$ calibrated parameters corresponds to the non-"state-space" GR4H version (not presented, seePerrin et al. (2003)), for which the calibration tool is provided. $x_4$ values in the present run with the "state-space" were set at 0.15.



*Code and data availability.*   The current DassFlow code is available on demand at https://www.math.univ-toulouse.fr/DassFlow/download.html.
The exact version of the model used in this article is archived on Zenodo (https://doi.org/10.5281/zenodo.6342723), as are input data needed
to run the main direct/inverse simulations.

*Author contributions.*   This work is part of the PhD work of LP under the supervision of PAG and JM. Research plan: PAG, LP, JM. Compu-
tational software DassFlow2D adaptation from previous version by LP. Numerical investigations by LP with analyses by all. JM has designed
or supervised the computational methods (1D2D numerical schemes and VDA algorithm). All authors have participated to the writing.

*Competing interests.*   The authors declare that they have no competing interests

*Acknowledgements.*   This study was rendered possible by the adaptation of covariance matrices in the inverse algorithm from the DassFlow1D
algorithm Larnier et al. (2020) by Lilian Villenave (INRAE Aix-en-Provence) and the implementation of final tricks in Pearl scripts for adjoint
post-processing in DassFlow by Kévin Larnier (CS Group Toulouse). The authors greatly acknowledge Pascal Finaud-Guyot (HydroSciences,
Univ. Montpellier) for fruitful discussions related to hydraulic solvers. They warmly thank Laurent Dieval from SPC-GAD France, the
operational flood forecasting service for the Gaves-Adour-Dordogne river networks, for providing data for the Adour case and also for
interesting discussions. They also thank Robert Mosé (ICUBE, Univ. of Strasbourg), the PhD director of LP.

*Financial support.*   PhD of LP has been co-funded by CNES and ICUBE. Project management: PAG.



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
