# Peer review of "Multi-dimensional hydrological-hydraulic model with variational data assimilation for river networks and floodplains"

_EGUsphere, 2022_

## Author Response (AR1)

**1 Reviewer n°1**

**1.1 General Comments**

The manuscript presents the development of a multi-dimensional hydrologicalhydraulic model. The model is calibrated/optimized using the variational data assimilation approach. Enhancing the computational efficiency of multi-dimensional river routing is important in the field of hydraulic modeling. Furthermore, it is important to investigation on state-parameter estimation utilizing the assimilation approach to identify the optimal parameters for obtaining better estimates of the physical variables of hydrodynamics. The authors describe a scientifically sound approach for performing multi-dimensional hydrologic-hydraulic modeling and calibrating model parameters using variational data assimilation. However, the manuscript lacks the reasoning and purpose for multi-dimensional hydrologichydraulic modeling over 2D modeling of the entire river length (a lot of commercial software are available for such 2D modelling).

On the other hand, it is unclear how the combination of VDA with multidimensional modeling will increase the capacities of estimating physical variables.

The use of several acronyms and mathematical formulae with no physical relevance impeded the intelligibility of the work considerably. Some specific and technical comments are provided to enhance the text that will be published in Geoscientific Model Development.

• We thank the reviewer for his careful and positive evaluation of our work and for providing helpful cristism. The reviewer brought up valid concerns about the clarity of the manuscript and explanations regarding the reasoning behind the proposed numerical tool. This lead to modifications of the introduction and parts of the main text which should solve the above remarks by better highlighting the novelty of the work compared to existing research work and software. The reviewers specific comments and technical correction are answered point by point below and in the article.

**1.2** Specific Comments**

- 1. In the introduction, the authors did not clearly mention the motivation and the objective of the study. The authors mentioned "high resolution accuracy and fast computation times" but they introduced many studies on the matter in the next paragraph (L70-79). Authors should present more focused science questions.
  - (a) Many clarifications were added regarding the context and motivations along with more clearly detailed scientific questions and methodological steps.
    - i. The general scientific challenges related to combining multi-source data and numerical models is described from line 45 to 54.

- ii. Advantages of the new model with regards to other large scale hydraulic-hydrological models are underlined form line 82 to 89.
- iii. Focused scientific issues are detailed from line 91 to 94, in complement to more technical objectives.
- iv. The assets and capabilities of the model are regrouped from line 95 to 114.
- 2. It is a bit confusing why the authors combined the multi-dimensional modelling with multi-source data assimilation methods over full 2-D modelling.
  - (a) The motivations and originality of multi-D modeling are now better explained, regarding full shallow water model application at large scale that is still difficult, but also for high dimensional spatio-temporal parameters inference with VDA method.
    - i. As demonstrated in the numerical results on comprehensive synthetic and real experiments, the 1Dlike modeling approach allows for the reproduction of fine scale spatial and temporal signatures (i.e. hydraulic controls and flood waves). It does so at a low computational cost (13 s per day on a single core for the Adour case with 180 km of total river length), which opens the way for its use over large scale river networks. This is compounded by the fact that it can be built from widely available global datasets (e.g. LiDAR DEMs or satellite imagery of river surfaces). The 1Dlike modeling strategy thus provides interesting and physically sound new modeling capabilities over full 2D modeling approaches, while still allowing for the modeling of fine 2D hydrodynamics at areas of interest thanks to its 1D2D interfaces.
    - ii. Furthermore, the new code is implemented within an integrated variational framework that is suited to solve both the hydraulic and hydrological inverse problems of high-dimension and over large domains (see answer to Comment 3 just below).
    - iii. Finally, note that the integration of a hydrological model in the variational assimilation toolchain itself is an innovation. It benefits from the lower computational cost of the 1D2D models, which allows simulation and inferences over larger time periods.
- 3. The authors should explain the reason for using variational data assimilation method over ensemble data assimilation methods.
  - (a) The VDA method is adapted to infer high dimensional parameter vectors of non linear hydrodynamic models, from heterogeneous data as shown by several works (incl. of the authors) on hydraulic parameter inference in sparse multi-satellite observation context. The following sentences were added in the reworked introduction regarding VDA: "(...) adequate method to tackle such high dimensional and multi-variate hydrodynamic inverse problems from heterogeneous datasets

(...)". "Let us remark that this is not the case for stochastic methods, where computational cost can increase exponentially with the number in sought parameters."

- 4. It is not clear from the text in which temporal scale the parameter optimization is performed ?
  - (a) All inference results of temporal parameters are obtained from observations of their full temporal scale. Precisions were added to all academic case, where this may have been ambiguous.
- 5. What are the difference between parameter optimization (i.e., VDA) between 1D and 2D cases.
  - (a) The difference lies in the forward model setup (1Dlike, 1Dlike-2D and 2D) and hence in the corresponding control vector, plus eventually the observations considered, but the VDA algorithm remains the same. For a given case, parameters can be inferred simultaneously, seamlessly and using the same observations and cost function. One goal of the presented work is to provide this simple integrated chain. Clarifications were added at line 260 : "Note that 2D, 1Dlike and 1D2D models are functionally identical from the point of view of the adjoint code and the assimilation process."
- 6. Can the methods shown in the manuscript be applied for all spatial scales? A discussion of the spatial resolutions 1D or 2D river segments are needed for fully utilization of the methods developed in this manuscript.
  - (a) The presented method is able to infer large numbers of parameters, uneven in space and time, over the whole assimilation time window, given sufficient data to constrain the inverse problem. The application of the method to varied spatial scales is contrained by data availability but not the method itself. Sentences added in conclusion: "The methods for direct and inverse modeling can indeed be applied at multiple spatial scales including with fine resolution (imposing finer calculation time steps to respect CFL condition in the forward model). Building and constraining the models is, however, dependant on data availability and the informative content of observations, which may be linked to the spatial scale."
- 7. In the synthetic experiment, the authors discuss only scenarios with no lateral flow (e.g., surface and subsurface runoff) but it is better to have some discussion with lateral flow case.
  - (a) The addition of lateral flows (either surface or subsurface in a weak coupling way) is possible via the imposition of boundary conditions, which is numerically identical to the addition of upstream flows,

which has been studied on the Adour basin. Furthermore, the assimilation of lateral flows with a VDA appraach over large rivers was studied in (Pujol et al., 2019). Sentences added in conclusion.

- 8. Many of the mathematical equations found in the main text are repeated in the appendices. So, I would like to suggest the authors to use the equations in the appendices to help them explain the main text more clearly. Authors can reduce the number of repeating equations by doing so.
  - (a) Thank you for this useful comment. The redundant equations were changed to better leverage the Appendix (eq.(4) was replaced by a general hydrological formulation instead of the specific GR4 equations featured in the Appendix).
- 9. Is it possible for the model to modify the dimensionality on a temporal scale? When a flood occurs in one river reach, the flood is simulated using a 2D mesh, while in other cases, a 1D method is employed. If this is true, how will the model determine the flooding times?
  - (a) If we understand correctly the first part of the question, the present model does not have dynamic mesh refinement features. However, the multi-D model enables to model the evoluiton of (h, u, v) variables of the full shallow water equations, and thus flow lines, flood propagation in the network and fine inundations dynamics over floodplains, hence flooding times, flood extents and times of local overflows. The cases with 1D approaches only are used for forward solver validations and also inference tests where 1D configuration is sufficient.
  - (b) A fully 1Dlike model, like that of the Adour case, could be used to simulated non-flooding events and trigger the run of finer 1D2D and 2D models in case of flood. At a given water level associated with flooding over an area of interest, the hydraulic states of the 1Dlike model could be used to initialize a 1D2D and/or full 2D model, in order to obtain flooding times. This kind of application, which may or may not be needed depending on specific operational modeling objectives, is left for further work.
- 10. In section 3.3.1, the authors present a observing system simulation experiment (twin experiment) where a virtual observation is assumed. When they used virtual observations to calibrate the model parameters, they assumed the observations are available in all the river pixels and all the time. The availability of observations for all the river reaches in all the time may not be reasonable. To assess the validity of methods the authors should test more realistic scenario by assuming either spatial or temporal discontinuity.
  - (a) The purpose of the calibration of the 1Dlike Garonne model using dense observations of a 2D reference model is to illustrate its capability to represent complex spatial variabilities. This inference from a

real fine model densely observed is sufficient to validate the pertinence of the 1Dlike model to reproduce real like physics (recall observation of a fine 2D model), as well as to validate the adaptation of the VDA algorithm for the hydro-au multi-D model (thanks to other inference cases also). Real and complex hydraulic inverse problems have been extensively studied by the authors, for example in sparse satellite data context (Brisset et al. 2018, Larnier et al. 2020, Pujol et al. 2020) with the VDA method on which is based the present one. This is now hopefully more clear in the manuscript, the following sentence was added : "This dense observability is not meant to imitate the actual observability of real rivers, but to provide sufficient constraints for the considered problem, with the aim to showcase the capability of the variational toolchain and 1Dlike model to fit fine scale real-like WSE variations (see hydraulic parameters inference from sparse altimetric data in Brisset et al. (2018); Larnier et al. (2020); Garambois et al. (2020); Pujol et al. (2020))."

**1.3** Technical Corrections**

- 1. L60: What does "precipiton" mean
  - (a) The sentence was divided in two to better introduce the concept of precipiton : "In Hocini et al. (2020), an original 2D hydraulic modeling approach is used to compute steady inundation maps of various return periods at high resolution (5 m) for river networks and floodplains at catchment scale of several thousands of square kilometers (up to 5050 km2). It uses "precipiton" for the resolution of the full shallow water model, which consists in propagating elementary water volumes on the water surface, as proposed by Davy et al. (2017)."
- 2. L73 It is not easy to guess local 2D 'zooms'. Please elaborate on it.
  - (a) Details were added to better define the method and its purpose: "or more recently by coupling 1.5D and 2D equations, so as to model local 2D zooms of overflows overlapping with a 1D domain for withinbanks flows, in a variational data assimilation framework, in Gejadze and Monnier (2007); Marin and Monnier (2009). "
- 3. L83 SWE is not defined before
  - (a) "SWE" is now properly introduced in this sentence.
- 4. Tabel 1: what is "sources available" better to explain it in the caption
  - (a) The term "sources available" was defined in the caption : " "Sources available" means that the source code is freely accessible, either through direct download or upon request."
- 5. L113: What do SW means, "shallow water"?

- (a) SW did indeed stand for "shallow water". It was replaced with the full term to avoid the excessive use of acronyms.
- 6. L129: [0, T] reads [0, T]
  - (a) The notation was rectified.
- 7. Eq (1): doesn't U, F, G, Sg, and Sf be introduced?
  - (a) Omitted notations were introduced: "U is a vector of state variables and F(U) (resp. G(U)) is its flux over x (resp. (y). Sg is a gravitational term and Sf is a friction term. "
- 8. Section 2.2.4 does not contain in any title
  - (a) The section head was a typo and was removed.
- 9. L201-204: do xi and t refers to location and time, respectively?
  - (a) The paragraph was reworked so this is clearer: "Over the hydrological domain  $\Omega_{rr}$ , we consider the discretization  $\mathcal{D}_{rr}$ , here into C subcatchments  $\{bv_1, ..., bv_C\}$  and corresponding disjoints sub-domains  $\Omega_{rr,i\in[1..C]} \subset \Omega_{rr}$ , which outlets coordinates are respectively  $x_i \in$  $\Omega_{rr}, i \in [1..C]$ .
- 10. L211: What are Qr and Qd
  - (a) Qr and Qd are now defined exclusively in the appendix and notation are consistent.
- 11. Eq(9): latter part of Eq(9) is missing.
  - (a) This formulation of the optimization problem was reworded to :

$$k^{\star} = \operatorname{argmin} J\left(k\right) \tag{1}$$

- 12. Figure 6: The authors can include the x, y axis in the panel (b). Is it possible to show an example of WSE variation at the 1D-2D mesh boundary? Display the legend in the center panel (blue/red lines on the hydrograph). This statement applies to all similar figures.
  - (a) (x,y) axes where included in the (b) panels of Figures 6 and 7. Both figures were updated to show a zoom of the waterline at the upstream 1D2D interfaces. Legends where modified according to reviewer comment.
- 13. L332: What is the reference discharge for calculation of NSE.
  - (a) The mean discharge at outlet for the varied, non-flooding flow event is  $398 \text{ m}^3/\text{s}$ . This information was added on the caption of Table 3.

- 14. Figure 12: Explain the area zoomed in caption, what are the two-river section shown in the focused area, explain them in the caption.
  - (a) The following description was added to the caption: "In the zoom, both models are represented with a slight longitudinal shift to allow qualitative result comparison of both simulated water depth and water surface extents."
- 15. L415: What is SMS meshing tool mean?
  - (a) A footnote was added: "https://www.aquaveo.com/software ; a software including comprehensive meshing tools"
- 16. L499: What is PET standard for?
  - (a) PET was replaced by its full form: "potential evapotranspiration"
- 17. L500: What is SMASH?
  - (a) The SMASH platform was better described: "Spatial averages of the rainfall and potential evapotranspiration computed using the SMASH distributed hydrological modeling platform (Jay-Allemand et al. (2020); Colleoni et al. (2021))

**2 Reviewer n°2**

This study proposes an integrated hydrological and multi-dimensional hydraulic modeling approach that is capable of handling multi-variate optimization problems of high dimension using multi-source data. The new multi-D hydraulic computational model was coupled to the formerly developed hydrological model (GR4H) in a semi-distributed setup, called DassFlow2D-V3. The topic is of high importance to the hydraulic and hydrology community, particularly under the massively growing high-resolution data or products obtained by remote sensing. However, the novelty of the work seems to be exaggerated. The basis of the proposed hydrologic-hydraulic model was already developed by Monnier et al. (2016) and Santos et al. (2018) and as the authors acknowledge, this work presents an upgrade to the above setting with respect to a new multi-dimensional hydraulic computational model.

Furthermore, although the authors have described the capabilities of similar models (L50-70), the need for and the striking advantage of the proposed framework over the competing models have not been demonstrated.

The manuscript also lack the underlying science questions that need to be outlined clearly. Further technical and editorial comments are listed below to consider before manuscript can be published in Geoscientific Model Development.

• We thank the reviewer for his evaluation and feedbacks on our work. A reworked introduction is proposed to address the main issues raised as well as other minor corrections through the draft, while better explaining the

scientific novelty of our work, i.e. a new multi-dimensional hydrologicalhydraulic computational model with VDA, as well as the required numerical model upgrades.

- The proposed approach enables to solve, with a unique numerical tool including state-of-the-art know-hows in numerical hydrology-hydraulics with a new multi-D solver and VDA, high-dimensional and difficult forward-inverse modeling problems, over entire catchments and river networks with floodplains and is adapted to ingest heterogeneous multi-source datasets. This new approach enables to tackle large computational domains with a full shallow water model, which is currently only performed with simplified models as pointed out in the introduction. Furthermore, the presented multi-D approach is a novelty that allows the seamless interfacing of 2D meshes for fine scale hydrodynamics and low computational cost 1Dlike reaches, which are shown to allow a good representation of spatial and temporal real-like signatures at the catchment scale. The striking advantage of this model over its competitors is the integration of the above capabilities in a single toolchain.
- Despite existing hydraulic and hydrological numerical codes from which this work started, the amount of work needed to reach the presented results is considerable and of relatively high technicity. It consisted mainly, but not only in code developments related to the implementation of:
  - \* The multi-D algorithm, via the modification of finite volume solvers.
  - \* Multi-D meshes and the internal coupling strategy adapted to them.
  - \* Hydraulic-hydrological coupling via boundary conditions.
  - \* GR4 state-space source code differentiation and validation, with full integration to the assimilation toolchain.
  - \* Regularization into the VDA algorithm.
  - \* Academic and real cases building with extended testing of the new forward-inverse model features.

**2.1 Major comments**

- 1. The material presented in section 2 is hard to follow in many parts: too many acronyms and multiple cross-references to other sections not presented, yet. Also, the figures are not referred to as they appear in the order presented. The authors are invited to carefully review the manuscript for a clearer and smoother presentation.
  - (a) Thank you for this advice.

- i. Unnecessary acronyms and equations have been removed or altered (see Minor comments).
- ii. Figure are now cross-referenced in order, as early cross-references to figures in the results section were replaced by cross-references to Subsections and some repetitive figure cross-references were removed.
- 2. L215-217: It seems only some parameters were calibrated in the integrated model, but the authors should describe the reason behind this choice.
  - (a) The calibrated parameters are the reservoir capacities, which are catchment dependent. The remaining "fixed" parameters are meant to represent hydrological processes and are assumed to not be catchment dependant in the GR4 approach (see Perrin et al., 2003). This information was added to section 2.3.
- 3. L240: Why did not you consider the square root of the current objective functions to make their unit tangible and comparable to the unit of the estimating variables, e.g., Q?
  - (a) The simulated and observed quantities are homogeneously compared with a difference inside the norm. This is a quadratic cost function, hence convex, as needed in VDA.
- 4. Section 2.4: It is not clear what the implication of "variational" is in the VDA framework.
  - (a) VDA stands for "variational data assimilation" methods using on a gradient-based algorithm to minimize a cost function; hence gradients of the cost function, i.e. partial derivatives, or in other words, by definition of derivation, rates of variation of the cost function w.r.t. the sought parameters. Parts of section 2.4 were reworked to make this information clearer. : "Given spatio-temporal flow observables, provided by in situ and airborne sensors for instance, the inverse algorithm consisting in Variational Data Assimilation (VDA) aims at estimating the unknown or uncertain "input parameters" of the hydrological- hydraulic chain composed of a hydraulic model, presented in Section 2.2, and a hydrological model, presented in Section 2.3. Detailed know-hows on VDA may be found in online courses (see e.g. Bouttier and Courtier (2002); Monnier (2014)). This group of algorithms uses cost gradients, i.e. variations, here computed by the adjoint model, to minimize a misfit to the observed reality (see Fig. 5). The adjoint model is obtained by automatic differentiation, using Tapenade Hascoet and Pascual (2013)."
- 5. L334-338: You have repeated this experiment setting for at least three times in the manuscript. The same issue is seen in other parts. The authors are highly recommended to avoid repeating the same material, but

with different toning, as well as to confine the results section to what are really the results. Currently, the results sections includes material related to the details of different experiments that should be stated in section 2.

- (a) Thank you for identifying this issue. Case description of synthetic models was improved and unceessary repetitions were removed.
  - i. In Subsection 3.1:
    - A. Inferences of temporal forcings (inflow hydrographs  $Q_i(t)$ ,  $i \in [1..N]$ ) are presented on a 1D2D confluence case. Inferences of channels parameters are presented on a straight 1Dlike case. Next, inferences of hydrological parameters  $(c_i)_{i \in 1..4}$  are presented using hydraulic observables." was replaced by "Inferences of temporal forcings, channel parameters and hydrological parameters are carried out." as details are present in the latter subsections.
    - B. Similarly, "inally, the model is tested on two real cases: (i) the capability of the 1Dlike model to reproduce real flow lines and propagations, through effective bathymetry-friction (b(x), n(x))" was replaced by "Then, the model is tested on two real cases. The 1Dlike model is tested against a reference 2D model built on fine bathymetry of 75 km of the Garonne river. The whole multi-D hydraulic-hydrological tool chain is tested on the Adour River network."
  - ii. In Subsection 3.2:
    - A. The following sentence was removed : "(i) a rectangular prismatic channel, (ii) a rectangular channel with a slope break and (iii) a parabolic prismatic channel. " as it is present in Subsection 3.2.2
- 6. I am maybe missing something, but from the results it looks like the proposed new modeling framework does not that remarkable advantage in comparison to the formerly developed models of the same purpose. The authors should clearly highlight the distinct advantages of the proposed framework based on the reported results.
  - (a) This remark is now addressed in the last part of the introduction (see answers to questions 1 and 2 for Reviewer n°1). The conclusion was also updated, notably by the addition of this sentence: "This new toolchain opens the way for the resolution of large scale, high dimensional inverse problems that can be considered given constraints from multi-source datasets.".
- 7. To evaluate the accuracy and efficiency of the proposed model, the authors should expand their test cased to real-world river basins of small of medium size (< 1,000 km2) and compare the reproduced hydrographs with observations at multiple points across the basin. Currently, it is really difficult to

judge about the applicability of the model as well as the relative advantages relative to the other competing models.

- (a) The accuracy and efficiency of the proposed model is validated on a significant range of academic cases with regards to a state-of-the-art 2D SW model (DassFlow2D from Monnier et al. 2016) achieving the best know-hows and accuracy in terms of numerical schemes accuracy and VDA capabilities. Moreover, it is successfully applied on two real cases in the present article. The 1Dlike approach was validated, in terms of propagations and spatial control reproduction, on the Garonne case. It was also applied on a relatively large and complex river basin, where simulated propagations times are satisfying and hydrographs are inferred from data at real-world observation points. For the purpose of this article, which is aimed at presenting new developments in an already numerically sound computational platform, the authors deemed these experiments sufficient. We believe that the material presented, over a range of academic and real cases is rigorous and highly sufficient to validate the proposed implementations as well as their pertinence for real cases. Multiplying real cases would be at the expense of rigorousness and implementation details which are important to us and perhaps for the readers of GMD. Moreover a detailed analysis of applications to a range of real-world river basins with multi-source datasets represents on its own a research study (or more!) and will be further studied.
- 8. The authors should also discuss the computational cost of the proposed model. Obviously, this issue would be interesting in case of real-world medium size basins, not the very simple virtual experiments considered in this study. In summary, the author should discuss if their model has the potential of application in the flood forecasting systems at a country or continent scale.
  - (a) On the Adour 1Dlike model, a 24h direct simulation takes less than 15 seconds on average (on a single thread). It could be used to quickly calibrate river network parameters, with the intent of using the calibrated parameters as input of the more computationally costly multi-D model.
    - i. The following paragraph was added in Section 4 : "For example, a 1Dlike model, with low computation cost, may be used to calibrated parameters with VDA over a large river networks, which would provide parameters to be used in a corresponding 1D2D model, aimed at modeling local flood dynamics."
    - ii. In Section 3.3.2, the following comment was added : "A 24h simulation runs in around 13s, using uncalibrated parameter estimates."

- (b) During a flooding event, the 1D2D model of the Adour networks simulates a day in around 6 hours (again, on a single thread). This information was added to Subsection 3.3.2).
  - i. This time could be reduced by reworking the 2D mesh to better leverage 1D2D capabilities, i.e. by modeling the main river channel in this area using 1Dlike cells. This would allow much lower computation times before and after the flood, as finer time steps needed to respect CFL condition in the forward hydraulic model would cover less of the total simulation time (see answer to question 6 of reviewer n°1). To do this, particular attention should be paid to the fine effective bathymetry at the 1D2D interface (i.e. the top of the levees/banks).
  - ii. Further methods to reduce computational costs while keeping the full advantages of the presented multi-D method could be developed in further work (see e.g. answer to question 9 of reviewer  $n^{1}$ ).
- (c) Investigations on a larger sample of real world cases, including varied scales from medium, to country-wide, to continental are worthy of a whole dedicated study or more.

**2.2 Minor comments**

- 1. Figure 1 needs improvement, and a reader cannot easily grasp the illustrated conceptual meshing approach.
  - (a) The graph was updated. It defines the hydraulic and hydrological domains more clearly, and reminds the reader of the assumptions used in the multi-D hydraulic domain. The caption was updated accordingly.
- 2. L7: virtual experiments instead of "academic"?
  - (a) Changed to virtual
- 3. L42: "a" key to...
  - (a) Changes were made according to reviewer suggestion to avoid confusion. "key to" is also a valid form with the same meaning, that does not imply exclusivity.
- 4. L84: SWE not defined before in text.
  - (a) "SWE" is now properly introduced in this sentence.
- 5. Table 1: The citations are numbered in this table, while they are not labeled in the reference list.
  - (a) Thank you for pointing out this issue. Citations style in the Table was set to (Author, YEAR) format.

6. L109: VDA not defined before.

(a) "VDA" is now properly introduced in this sentence.

7. Section 2.2.4 lacks a title.

(a) The section head was a typo and was removed.

- 8. L230: the aim here is to ... please fix. The manuscript needs several other English proofing instances that the authors should take care of them.
  - (a) The sentence is rectified.
- 9. Figure 6: What is the blue time series in the second panel?
  - (a) Legends for figures 6 and 7 were updated.